# DISCOVERING LIE GROUPS WITH FLOW MATCHING

## ABSTRACT

Symmetry is fundamental to understanding physical systems, and at the same time, can improve performance and sample efficiency in machine learning. Both pursuits require knowledge of the underlying symmetries in data. To address this, we propose learning symmetries directly from data via flow matching on Lie groups. We formulate symmetry discovery as learning a distribution over a larger hypothesis group, such that the learned distribution matches the symmetries observed in data. Relative to previous works, our method, `LieFlow`, is more flexible in terms of the types of groups it can discover and requires fewer assumptions. Experiments on 2D and 3D point clouds demonstrate the successful discovery of discrete groups, including reflections by flow matching over the complex domain. We identify a key challenge where the symmetric arrangement of the target modes causes "last-minute convergence," where samples remain stationary until relatively late in the flow, and introduce a novel interpolation scheme for flow matching for symmetry discovery.

## 1 INTRODUCTION

Symmetry has been central to mathematics and physics for over a century, and more recently, has featured prominently within machine learning. Classically, the underlying symmetry is first identified and categorized; Noether's theorem (Noether, 1918) establishes the existence of corresponding conservation laws, which can be used to model and understand physical systems. In machine learning, symmetries serve as powerful inductive biases in equivariant neural networks (Cohen & Welling, 2016; Kondor & Trivedi, 2018; Weiler & Cesa, 2019). These architectures leverage known symmetries in the data, such as rotational invariance in molecular structures (Thomas et al., 2018; Satorras et al., 2021) or translational equivariance in images (LeCun et al., 1998), to achieve superior performance with fewer parameters and training samples (Bronstein et al., 2021).

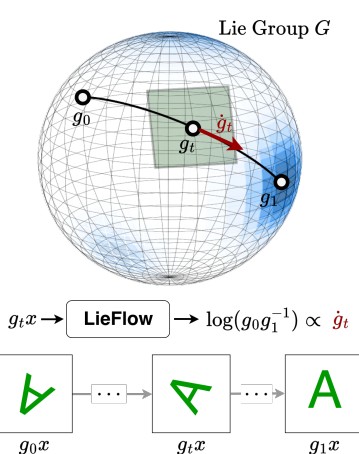

Figure 1: Flow matching for symmetry discovery

A critical limitation in both physics and machine learning is that the exact symmetry group must be known a priori. In practice, the underlying symmetries are often unknown, approximate, or domain-specific. In physics, many systems exhibit hidden symmetries, from a simple 1D harmonic oscillator to black holes (Gross, 1996; Liu & Tegmark, 2022). In material chemistry, non-obvious symmetries such as valency and electronegativity can reduce the combinatorial search space (Davies et al., 2016). In computer vision, objects often contain local symmetries, and real-world scans are often partial or noisy (Mitra et al., 2006; 2013). These applications share a common need: the ability to automatically discover and characterize symmetries from data, without relying on domain expertise or manual specification.

Several prior works have focused on symmetry discovery in restricted settings, such as roto-translations in images (Rao & Ruderman, 1998; Miao & Rao, 2007), commutative Lie groups (Cohen & Welling, 2014), or finite groups (Zhou et al., 2020; Karjol et al., 2024). Recent studies on learning Lie groups also have some limitations: Benton et al. (2020) assume a fixed Lie algebra basis and a uniform distribution over the coefficients; Dehmamy et al. (2021) produce non-interpretable

symmetries; Yang et al. (2023) assume a Gaussian distribution over the Lie algebra coefficients and use potentially unstable adversarial training; and Allingham et al. (2024) use a fixed Lie algebra basis and evaluate only over images.

In this work, we propose LieFlow, which discovers symmetries via flow matching directly on Lie groups. We formulate symmetry discovery as learning a distribution over a larger symmetry group, which serves as a hypothesis space of symmetries, and the support of this distribution corresponds to the actual transformations observed in the data. Using flow matching allows us to capture both continuous and discrete symmetries within a unified framework and accurately model the highly multi-modal nature of symmetries. Unlike standard flow matching (Lipman et al., 2023), which operates in data space, LieFlow learns flows directly over Lie group manifolds, conditioned on the data samples, enabling the generation of new plausible data that respect the underlying symmetry structure.

Our key contributions are to:

- formulate symmetry discovery as a flow matching problem on Lie groups to match the true data symmetries,
- provide a unified framework to discover continuous and discrete symmetries, including reflections via flow matching over the complex domain,
- identify the "last-minute mode convergence" phenomenon, causing near-zero vector fields for most timesteps, and introducing a novel time schedule for symmetry discovery.

## 2 PRELIMINARIES

**Lie Group** A Lie group is a group that is also a smooth manifold, such that group multiplication and taking inverses are also differentiable. This makes them particularly suitable for describing continuous symmetries. For example, the group of planar rotations $SO(2)$ is a Lie group consisting of one parameter $\theta$, where the rotation by angle $\theta$ can be represented as $R_\theta = \left[\begin{smallmatrix} \cos\theta & -\sin\theta \\ \sin\theta & \cos\theta \end{smallmatrix}\right]$. Each Lie group $G$ is associated with a Lie algebra, denoted as $\mathfrak{g}$, which is the vector space tangent at the identity. The Lie algebra captures the local (infinitesimal) group structure and describes how the group behaves near the identity.

The exponential map $\exp : \mathfrak{g} \to G$ defines the relationship between the Lie algebra and the Lie group. For matrix Lie groups, which we consider exclusively in this work, the exponential map is the matrix exponential $\exp(A) = \sum_{k=0}^{\infty} \frac{A^k}{k!}$. If the exponential map is surjective, we can restrict its domain and define the logarithm map $\log : G \supset U \to \mathfrak{g}$ for the neighborhood $U$ around the identity such that $\exp(\log(g)) = g, \forall g \in U$. Note that any Lie algebra element $A$ can be written as a linear combination of the Lie algebra basis, and the group element $\exp(A)$ is generated by exponentiating the linear combination. This allows us to generate the entire connected component of the Lie group using the Lie algebra, and thus the Lie algebra basis $L_i \in \mathfrak{g}$ is also called the (infinitesimal) generators of the Lie group.

**Flow Matching** Flow Matching (FM) (Lipman et al., 2023; Liu et al., 2022; Albergo et al., 2023) is a scalable method for training continuous normalizing flows. The goal is to transport samples $x_0 \sim p_0$ drawn from a prior (e.g., Gaussian noise) to data samples $x_1 \sim p_1$. This transport is described by a time-dependent *flow* [1], a family of maps $\psi_t : \mathbb{R}^D \to \mathbb{R}^D$ for $t \in [0, 1]$, where $\psi_t := \psi(t, x)$ and $\psi_0(x_0) = x_0$ and $\psi_1(x_0) \approx x_1$. This flow is defined through the ODE $\frac{d}{dt}\psi_t(x) = u_t(\psi_t(x))$, where $u_t$ is a time-dependent velocity field governing the dynamics. To construct a training signal, FM interpolates between $x_0$ and $x_1$ over time, e.g., through the straight-line interpolation

$$x_t = (1-t)x_0 + tx_1, \quad \dot{x}_t = x_1 - x_0, \tag{1}$$

defining a stochastic process $\{x_t\}$, also referred to as a *probability path*.

In Flow Matching, we define the *velocity field* $u_t$ that generates the desired probability path, and train a neural velocity field $v_t^\theta$ to match it:

$$\mathcal{L}_{\text{FM}}(\theta) = \mathbb{E}_{t,x} \|v_t^\theta(x) - u_t(x)\|^2. \tag{2}$$

---

[1]$\psi_t$ is technically an evolution operator arising from a time-dependent vector field; we follow standard terminology in the flow matching literature and use the term flow.

Since $u_t$ can be written as the conditional expectation of these per-sample derivatives, $u_t(x) = \mathbb{E}[\dot{x}_t \mid x_t = x]$, we obtain the practical *Conditional Flow Matching* (CFM) loss

$$\mathcal{L}_{\text{CFM}}(\theta) = \mathbb{E}_{t,x_0,x_1} \|v_t^\theta(x_t) - \dot{x}_t\|^2, \tag{3}$$

where $x_t$ and $\dot{x}_t$ come directly from the chosen interpolation.

This highlights the key advantage of FM: the transport dynamics can be learned entirely in a *self-supervised* manner, since interpolations between noise and data provide training targets without requiring access to likelihoods or score functions.

## 3 RELATED WORK

**Symmetry Discovery**  Many works on symmetry discovery differ on the types of symmetries that they can learn. Early works (Rao & Ruderman, 1998; Miao & Rao, 2007) used sequences of transformed images to learn Lie groups in an unsupervised way, but were limited to 2D roto-translations. Some recent works such as (Zhou et al., 2020; Karjol et al., 2024) can only discover finite groups, while others learn subsets of known groups (Benton et al., 2020; Romero & Lohit, 2022; Chatzipantazis & Pertigkiozoglou, 2023). LieFlow is flexible enough to learn both continuous and discrete symmetries and highly multi-modal distributions. Other related works focus on continuous Lie groups, as in our work, but suffer from other limitations. Dehmamy et al. (2021) uses Lie algebras in CNNs to discover continuous symmetries, but learn the symmetries end to end along with the task function, leading to non-interpretable symmetries. Moskalev et al. (2022) extracts Lie group generators from a neural network to analyze the learned equivariance, which is different to our objective of learning which symmetries exist in the dataset. Yang et al. (2023) use GANs to learn the Lie algebra basis, but assume a known distribution, often Gaussian, over these elements. Otto et al. (2023) provide a unified framework based on the Lie derivative to discover symmetries of learned models via linear algebraic operators, whereas LieFlow focuses on learning distributions over group transformations from data. Shaw et al. (2024) propose a method based on vector fields that discovers continuous and discrete symmetries of machine learning functions beyond affine transformations, focusing on Killing type generators rather than probability flows on a Lie group as in LieFlow. Ko et al. (2024) learn infinitesimal generators of continuous symmetries from data via Neural ODEs and a task-specific validity score, capturing non-affine and approximate symmetries, whereas LieFlow models a full distribution over transformations within a prescribed matrix Lie group.

The closest to our method is Allingham et al. (2024), where they first learn a canonicalization function in a self-supervised fashion and then learn a generative model that outputs the transformations for each prototype, given a large prior group. While the objective of learning a distribution over symmetries is similar, Allingham et al. (2024) is a two stage optimization process and uses costly maximum likelihood training. They also only evaluate on images, while we consider point clouds.

**Generative Models on Manifolds**  Several recent works have extended CNFs to manifolds (Gemici et al., 2016; Mathieu & Nickel, 2020; Falorsi, 2021) but rely on computationally expensive likelihood-based training. Some works (Rozen et al., 2021; Ben-Hamu et al., 2022; Chen & Lipman, 2024) proposed simulation-free training methods, in particular, Chen & Lipman (2024) consider flow matching on general Riemannian manifolds. However, none of these works specifically consider Lie groups or the task of discovering symmetries from data. Other flow matching works directly incorporate symmetries (Klein et al., 2023; Song et al., 2023; Bose et al., 2023), but do not discover symmetry from data. Other works consider score matching on Lie groups (Zhu et al., 2025; Bertolini et al., 2025), but again assume a priori knowledge of the group. Closely related to our work is Sherry & Smets (2025), who define flow matching specifically for Lie groups. While methodologically similar to our work, they assume prior knowledge of the symmetry group and use specialized implementations for different Lie groups.

## 4 LIEFLOW: SYMMETRY DISCOVERY VIA FLOW MATCHING

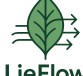

**LieFlow**

### 4.1 MOTIVATION AND FORMALISM

A central challenge in symmetry discovery is to identify the subset of transformations actually expressed in data without assuming explicit prototypes or fixed augmentation rules. Existing methods

often impose such structures externally, limiting scalability and generality. We address this by casting symmetry discovery as flow matching on Lie groups, which allows distributions over transformations to be learned directly from data while respecting the geometry of the underlying group.

**Problem Formulation.** Our goal is to recover the unknown group of transformations $H$ which preserve the data distribution $q$ on data space $\mathcal{X}$. To discover the hidden group $H$, we first assume a large hypothesis group $G$ which acts on $\mathcal{X}$ that contains the most common symmetries. We wish to find the stabilizer subgroup $H \subset G$ such that $q(hx) = q(x)$ for all $x \in \mathcal{X}$ and $h \in H$.

**Discovering Symmetry as Distribution Learning.** We frame the problem of finding $H$ as learning a distribution over the hypothesis group $G$ that concentrates on the subgroup $H$. To achieve this we train an FM model which flows a chosen prior distribution over $G$ to a distribution $p_\theta(g)$ supported on $H$. The structure of the discovered symmetries emerges from the concentration pattern of $p_\theta(g)$: continuous symmetries maintain a spread distribution across the manifold of admissible transformations, while discrete subgroups exhibit sharp peaks at a finite set of modes (e.g. the four rotations of $C_4$). Thus, `LieFlow` effectively filters the large hypothesis group down to the subgroup $H \subseteq G$ consistent with the data.

**Hypothesis Group and Prior over Symmetries.** We assume that the hypothesis group $G$ is connected, ensuring a well-defined logarithm map for compact groups. For the prior distribution $p$ over the group $G$, we use a uniform prior for compact groups and Gaussian priors over the Lie algebra coefficients for non-compact groups. When the exponential map is not surjective, we define the prior implicitly by sampling from the Lie algebra $\mathfrak{g}$ and applying the exponential map to construct group elements, ensuring well-defined training targets. In experiments, we consider $G = \mathrm{SO}(2)$ (planar rotations), $\mathrm{GL}(2)$ (invertible matrices) for the 2D datasets and $G = \mathrm{SO}(3)$ (3D rotations) for the 3D datasets. Although $\mathrm{SO}(d)$ and $\mathrm{GL}(d)$ for larger $d$ are certainly feasible in our framework, scaling our method to large or non-compact groups introduces additional numerical challenges, which we leave for future work.

**Interpolant Paths Along Orbits.** Unlike either traditional FM models which operate in the Euclidean data space or Riemannian flows which operate entirely on $G$, we propose an FM model that is conditioned on data but outputs group transformations. More precisely, consider a data sample $x_1 \sim q$ and a transformation $g \sim p(G)$ from the prior distribution. Let $x_0 = gx_1$. We wish to learn to flow from $x_0$ to $x_1$ using only transformations in $G$. Let $A = \log(g^{-1}) \in \mathfrak{g}$. We define the interpolation in the data space (an exponential curve) via the group action generated by $A$, where:

$$x_t = \exp(tA)x_0, \quad t \in [0,1], \tag{4}$$

This yields a probability path $p_t(x_t|x_0)$ in data space that stays on the group orbit of $x_1$.

**The Target Velocity Field.** The derivative of the path $\frac{d}{dt}x_t \in T_{x_t}\mathcal{X}$ defines a vector field along $x_t$. This defines a flow over $\mathcal{X}$ which stays within the orbit of $x_1$, only moving in directions dictated by the group action. To achieve symmetry discovery, we must learn both the flow from $x_0$ to $x_1$ and reconstruct the transformation $g$ that generated it. For convenience, we configure the flow network to output the Lie algebra element $A$ instead of $\frac{d}{dt}x_t$. We can then update $x_t$ by scaling and exponentiating $A$ and then applying the group action as in Algorithm 2. This allows us to reconstruct $g = \exp(-A)$. That is, we set $u_t(x_t \mid x_1) = A$. (Note that $A$ pushes forward to the velocity vector $\frac{d}{dt}x_t$ under the group action. See Appendix A for a more detailed derivation.)

**Objective.** The CFM objective on Lie groups is similar to the Euclidean case and is given by

$$\mathcal{L}_{\mathrm{LieCFM}}(\theta) = \mathbb{E}_{t, x_1 \sim q, x_t \sim p_t(x_t|x_1)} \| v_t^\theta(x_t) - u_t(x_t \mid x_1) \|_{\mathcal{G}}^2, \tag{5}$$

where $x_t$ follows the exponential curve defined previously, $p_t(x_t \mid x_1)$ is the distribution of points along the exponential curves, and $\mathcal{G}$ is a Riemannian metric on the Lie group, which equips the manifold with an inner product on each tangent space, allowing us to measure distances along the manifold. We use a left-invariant metric as it is completely determined by the inner product on the Lie algebra $\mathfrak{g}$ and the push-forward of the left action is an isometry, making computations simpler. The specific form of this metric is determined by the structure of the Lie group under consideration.

## 4.2 Training and Generation with LieFlow

**Training.** Using exponential curves and the flow matching objective, we can now derive the training and sampling algorithms for our method. The training algorithm (Algorithm 1) directly implements the Lie group flow matching objective by constructing training pairs $(x_0, x_1)$ where $x_1$ is sampled from the data distribution and $x_0 = gx_1$ for a randomly sampled transformation $g \in G$. $x_0$ is now a sample from the prior and lives in the group $G$. Unlike standard conditional flow matching, we construct $x_0$ from $x_1$ using $g$ and learn the flow over the Lie algebra of group $G$. As $G$ is a group, we can compute the target vector field $A$ analytically using the sampled group element $g$. We sample a timestep $0 \leq t \leq 1$ and construct a point $x_t = \exp(tA)x_0$ on the curve. Given $x_t$, the model predicts $A$. This means that the target vector field for this sample pair $(x_0, x_1)$ is constant over time. Thus the network only needs to learn to predict the Lie algebra element that generates the transformation from $x_0$ to $x_1$.

| **Algorithm 1** Training |
| --- |
| 1: **repeat** |
| 2:     $x_1 \sim q$ |
| 3:     $g \sim p(G)$ |
| 4:     $t \sim \mathcal{U}(0, 1)$ |
| 5:     $x_0 = gx_1$ |
| 6:     $A = \log(g^{-1})$ |
| 7:     $x_t = \exp(tA)x_0$ |
| 8:     Take gradient descent step on |
| 9:         $\nabla_\theta \|v_\theta(x_t, t) - A\|^2$ |
| 10: **until** converged |

| **Algorithm 2** Generating data |
| --- |
| **Require:** $x_1 \sim q$ |
| 1: $g \sim p(G)$ |
| 2: $x_0 = gx_1$ |
| 3: $\Delta_t = 1/T$ |
| 4: $M = I$ |
| 5: **for** $t = \{0, \frac{1}{T}, \frac{2}{T}, \ldots, 1\}$ **do** |
| 6:     $A_t = v^\theta(x_t, t)$ |
| 7:     $x_{t+\Delta_t} = \exp(\Delta_t A_t)x_t$ |
| 8:     $M = \exp(\Delta_t A_t)M$    ▷ Accumulate transforms |
| 9: **end for** |
| 10: **return** $x_1', M$ |

**Sampling and Generation.** During sampling (Algorithm 2), we integrate over the learned vector field to produce a new sample $x_1'$. The learned vector field predicts Lie algebra elements that are integrated using Euler's method with the exponential map $x_{t+\Delta t} = x_t \exp(\Delta t A_t)$, ensuring that the trajectory remains on the Lie group manifold throughout generation. We also output the composed transform $M$, where $x_1' = Mx_0$ as it is required for generating new group elements.

Algorithm 3 generates group elements $h$ consistent with the target group $H$. Given a data sample $x_1$, we transform it by $g \sim p(G)$ to obtain $x_0 = gx_1$. We then run Alg. 2 partially (lines 3-11) on $x_0$, obtaining outputs $x_1'$ and $M$. The composed transform $Mg$ forms a group element $h \in H$ that maps $x_1$ to $\hat{x}_1$. Note that the new sample $x_1'$ does not always match $x_1$ and in fact converges to the closest mode in the orbit (see Section 5.4 for more details), and thus this procedure does not always produce the identity element.

| **Algorithm 3** Generating Group Elements |
| --- |
| 1: $x_1 \sim q$ |
| 2: $g \sim p(G)$ |
| 3: $x_0 = gx_1$ |
| 4: $x_1', M$ from Alg. 2 Lines 3-11 |
| 5: **return** $Mg$ |

**Comparison with Standard Flow Matching.** While standard flow matching operates in Euclidean space, our approach operates on Lie group manifolds, learning vector fields in the tangent space (Lie algebra) that generate curved trajectories respecting the group structure. The exponential and logarithm maps replace Euclidean interpolation with geodesic-like paths on the group manifold. More importantly, we learn a generative model of transformations rather than data points, yielding a lower-dimensional problem on the group manifold itself, not over the entire data space. We formulate symmetry discovery as flowing from a broad hypothesis group to the specific symmetries in the dataset, with challenges analyzed in Section 5.4.

## 5 Experiments

To evaluate LieFlow, we generate several datasets with known symmetries of simple 2D and 3D point clouds of canonical objects (Figure 9 in Appendix B). For the 2D datasets, we consider the target groups $H = C_4, D_4$ and perform flow matching from $\text{SO}(2) \rightarrow C_4$, $\text{GL}(2, \mathbb{R})^+ \rightarrow C_4$, and $\text{GL}(2, \mathbb{C}) \rightarrow D_4$. To learn $D_4$ symmetries, we perform flow matching over the complex domain,

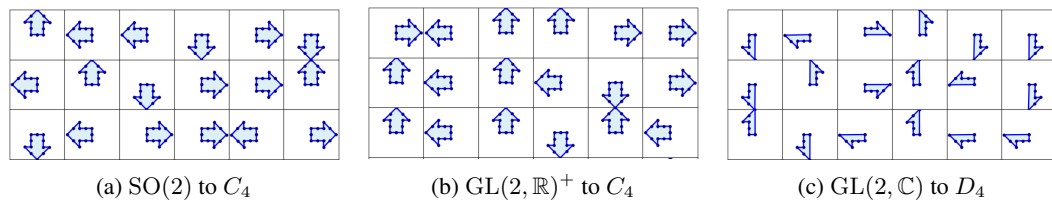

(a) SO(2) to $C_4$     (b) GL(2, $\mathbb{R}$)$^+$ to $C_4$     (c) GL(2, $\mathbb{C}$) to $D_4$

Figure 2: 2D Datasets: Generated data samples. The samples match closely to the original dataset symmetries of $C_4$ or $D_4$.

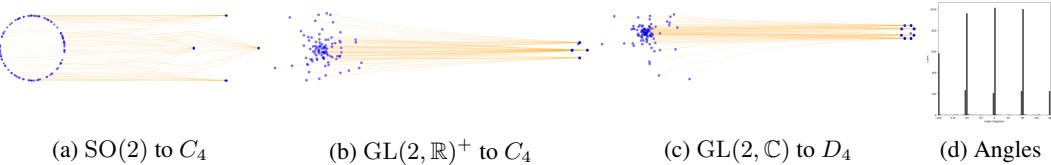

(a) SO(2) to $C_4$   (b) GL(2, $\mathbb{R}$)$^+$ to $C_4$   (c) GL(2, $\mathbb{C}$) to $D_4$   (d) Angles

Figure 3: 2D Datasets: Visualization of trajectories from $t = 0$ (left) to $t = 1$ (right) of the centroids of the transformed objects over 100 samples, and a histogram of learned angles for SO(2) to $C_4$.

where the data samples are first converted into complex numbers and passed through the complex-valued network, which uses twice as many neurons. We find that 20 steps for inference works well. For the 3D datasets, we use an object with no self-symmetries and consider the tetrahedral Tet, octahedral Oct and rotation over $z$ axis SO(2) as target groups. We consider flow matching from SO(3) → Tet, SO(3) → Oct, SO(3) → SO(2) and use 100 steps at inference. To quantify and comapre the quality of the learned symmetries, we compute the Wasserstein-1 distance between the generated group elements and the ground truth group elements. For all experiments, we use a simple 3-layer MLP with GELU (Hendrycks & Gimpel, 2016) activations. See Appendix B for more details.

## 5.1  2D DATASETS

Figure 2 shows generated samples from random $x_1$ in the test set. We see that most samples correctly align with the target distributions $C_4$ or $D_4$. Surprisingly, we find that flow matching in the complex domain, from $GL(2, \mathbb{C})$ to $D_4$, correctly recovers all 8 group elements. Considering that we are flowing from continuous groups to discrete groups, our method works quite well, with little variance at the target modes.

Figure 3 visualizes the centroids of a 100 generated samples over $t \in [0, 1]$. The $x$ coordinate of the centroids are offset as $t$ increases. As expected, we can see the centroids converge to 4 modes for the $C_4$ target groups and 8 for the $D_4$ group as $t \to 1$. These figures show that the learned distribution over the prior group becomes quite peaky, suggesting that it does indeed learn a discrete group quite well and learns the correct modes. Specifically for SO(2), we can decompose the learned transformation matrices (which should be very close to elements of $C_4$) and plot a histogram of the angles. As we have the ground truth transforms of the dataset, we first canonicalize $M$ in Alg. 2 and perform polar decomposition over it to obtain the rotation matrix and finally convert them to angles. Figure 3d shows that the learned distribution gives the correct peaks at the $C_4$ elements and is close to the correct distribution (mixture of 4 delta distributions).

Figures 4a, 4b show the intermediate timesteps during sample generation for SO(2) → $C_4$ and GL(2, $\mathbb{C}$) → $D_4$, respectively (GL(2, $\mathbb{R}$)$^+$ → $C_4$ is given in Appendix C.1). Interestingly, we see that depending on the random transformation to produce $x_0$ (line 3 in Alg. 2), the final generated sample does not match the original $x_1$ and seems to flow to the closest group element in the orbit of $x_1$ of the target group, i.e., $\arg\min_{gx_1}\{\|gx_1 - x_0\| : g \in G\}$. As the dataset contains all $C_4$ or $D_4$-transformed samples of the object, the flow still produces samples that are close to the data distribution. Furthermore, the progression shows that the vector field $v_t$ is close to 0 for the earlier timesteps (the determinants are close to 1) and only induces a flow towards the closest $gx_1, g \in T$ when $t$ is close to 1. We analyze this "last-minute mode convergence" phenomenon in Section 5.4.

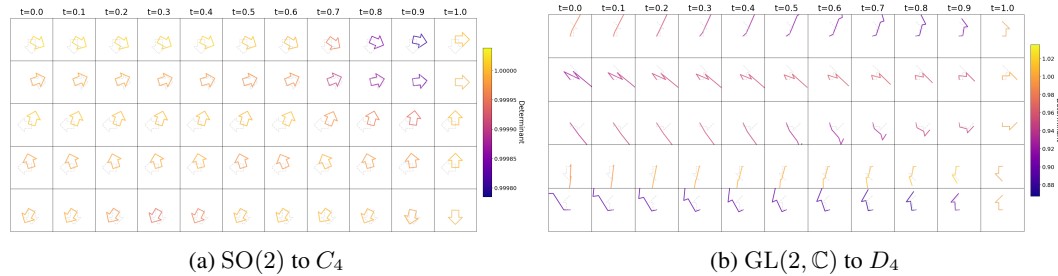

| | (a) SO(2) to $C_4$ | | (b) GL(2, $\mathbb{C}$) to $D_4$ |

Figure 4: Time progression of $x_t$ when generating 5 samples over 20 steps. The gray arrow shows the original $x_1$ and the color represents the determinant of the generated transformation matrix.

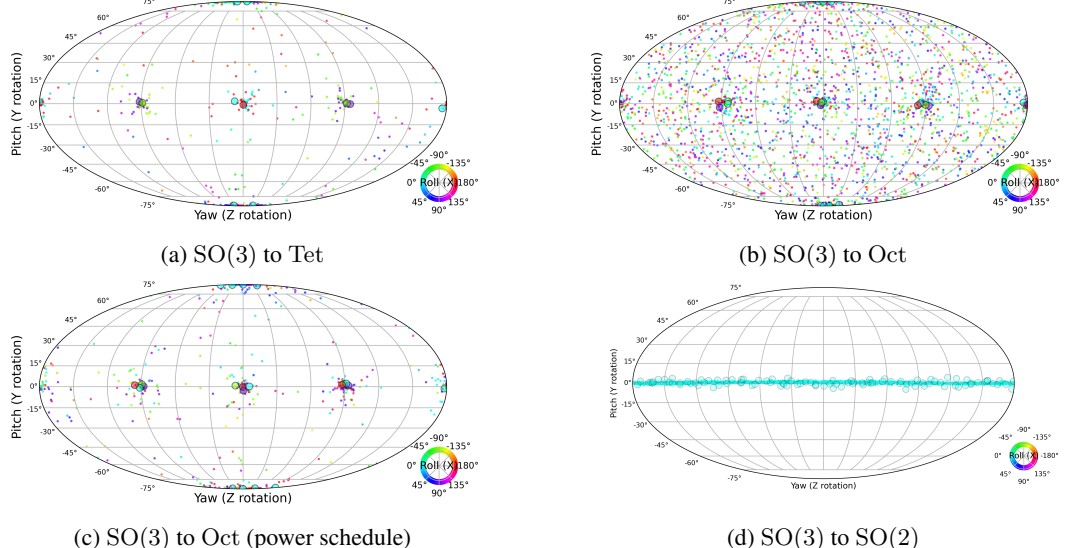

(a) SO(3) to Tet

(b) SO(3) to Oct

(c) SO(3) to Oct (power schedule)

(d) SO(3) to SO(2)

Figure 5: 3D Datasets: Visualization of 5,000 generated elements of SO(3) by converting them to Euler angles. The first two angles are represented spatially on the sphere using Mollweide projection and the color represents the third angle. The elements are canonicalized by the original random transformation and the ground truth elements of the target group are shown in circles with black borders. The points are jittered with uniformly random noise to prevent overlapping.

## 5.2 3D DATASETS

Figure 5 visualizes generated samples after training as points on SO(3) and the ground truth symmetries are shown with transparent circles with black borders. For the tetrahedral group (Figure 5(a)), our model learns to correctly output transformations close to Tet elements, with some errors. For the octahedral group (Figure 5(b)), our flow model is unable to extract the correct symmetries and learns virtually a uniformly random distribution over the prior SO(3). For SO(2) (Figure 5(d)), our model correctly learns to output rotations around the $z$-axis, with some small errors. Figure 12 (Appendix C.2) shows visualizations analogous to Figure 3, where our model seems to learn Tet and SO(2) elements, creating some visible clusters, but fails to discover Oct symmetries.

We also generate transformed point clouds given by our FM model for Tet and SO(2), as shown in Figure 13 (Appendix C.2). We can see several repeated point clouds, and see that some samples are rotations or reflections of other samples. We also show the intermediate $x_t$ during generation in Figure 16 and Figure 17 (Appendix C.2). For Tet group, as in the 2D case, we see that the transformed point clouds converge to the closest group element in the orbit. For the SO(2) group, we can see that the model produces rotations to turn one node of the tetrahedron to be aligned with the $z$-axis and one triangle face to be in the $xy$-plane, which corresponding to SO(2) symmetries around the $z$-axis. For the Oct group, we show its generated point clouds and the intermediate $x_t$ in Appendix C.2, where we can see that the FM model produces outputs close to the identity at most steps and fails to properly discover Oct.

Table 1: Wasserstein-1 distance between generated group elements and ground truth group elements in 2D experiments (lower is better).

| Method | $SO(2) \to C_4$ | $GL(2, \mathbb{R})^+ \to C_4$ | $GL(2, \mathbb{C}) \to D_4$ |
|---|---|---|---|
| LieGAN (1 gen.) | 1.688 | 1.688 | 1.803 |
| LieGAN (2 gen.) | 1.565 | 1.565 | 1.701 |
| LieGAN (4 gen.) | 1.672 | 1.672 | 1.131 |
| LieFlow(Ours) | **0.072** | **1.321** | **1.093** |

Table 2: Wasserstein-1 distance between generated group elements and ground truth group elements in 3D experiments (lower is better).

| Method | $SO(3) \to$ Tet | $SO(3) \to$ Oct | $SO(3) \to SO(2)$ |
|---|---|---|---|
| LieGAN (1 gen.) | 2.320 | 2.303 | 0.0975 |
| LieGAN (3 gen.) | 2.326 | 2.198 | 0.0693 |
| LieGAN (6 gen.) | 2.105 | 1.682 | 0.0528 |
| LieFlow(Ours) | **0.0885** | **0.0962** | **0.0429** |

Motivated by our analysis on "last-minute mode convergence" in the next subsection, where we observe that the entropy of the posterior remains stationary until large times, we introduce a novel time scheduling scheme to sample more timesteps near $t = 1$. Instead of sampling time from a uniform distribution $\mathcal{U}(0, 1)$, we propose using a power distribution where the density is computed as $p(x) = nx^{n-1}, x \sim \mathcal{U}(0, 1)$ and $n$ is the skewness parameter. The effect of different skewness values on the density is shown in Figure 19 (Appendix C.3). In our experiments, we chose $n = 5$.

In Figure 5(c), our modified time schedule now enables our model, whose network is unchanged from before, to correctly learn the octahedral group. Figure 21a (Appendix C.3) shows that our model correctly distributes probability mass towards the $24$ Oct group elements. To test the limits of our time schedule, we also consider the icosahedral group, with $60$ elements. However, our method fails in this case, where the model seems to output the identity for all inputs (see Figure 21b in Appendix C.3). It is clear that while our time schedule can alleviate the last-minute mode convergence issue and allow our method to discover more complex groups, it still cannot handle more "symmetric groups", i.e., higher-order subgroups.

### 5.3 QUANTIFYING DISCOVERED SYMMETRIES

To quantify the quality of the discovered symmetries, we compute the Wasserstein-1 distance between the empirical distributions of the generated group elements and the ground truth group elements in the test set consist of $5,000$ samples. We choose LieGAN (Yang et al., 2023) as our baseline since it is a recent generative approach that explicitly learns Lie algebra parameters for group-like transformations. For 2D datasets, we set the number of Lie generators to be 1, 2, and 4. For 3D datasets, we set the number of Lie generators to be 1, 3, and 6. The results are shown in Table 1 and Table 2. We can see that our method outperforms LieGAN in all experiments, demonstrating the effectiveness of our flow matching approach for symmetry discovery.

### 5.4 ANALYSIS ON LAST-MINUTE MODE CONVERGENCE

Given the challenges in learning the octahedral group, we hypothesize that using flow matching for symmetry discovery is in fact quite a difficult task. We illustrate with an even simpler scenario of flow matching directly on group elements (matrices), from $SO(2)$ to $C_4$. As the target vector field $u_t$ and the probability path $p_t(x_t \mid x_1)$ are important quantities in our FM objective (equation 5), we visualize them for a single sample in Figure 6.

Each target mode $x_1$ is colored differently and the velocities (group difference) to each $C_4$ group element are shown with the colored arrows. Near $t = 0$, the probability path $p_t$ is near uniform as $x_0$ is essentially noise and can be far away from $x_1$, i.e. there is still a lot of time $t$ for it to move to any mode. As $p_t$ is nearly uniform, the average of the velocities dominate and pulls $x_t$ close to the middle of the red and blue modes. Here, the vector field produces nearly $0$ mean velocity, as it is equidistant to the red and blue modes, and equidistant to the green and yellow modes. The distribution $p_t$ becomes peakier towards the closer modes (red and blue), but still remains somewhat

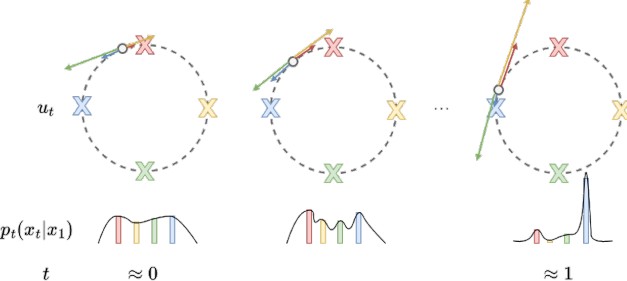

Figure 6: $\mathrm{SO}(2)$ to $C_4$ over group elements: target vector field and probability paths.

uniform. As such, the training target averages out close to $0$. Near the end of training, when $t$ is close to $1$, here the velocities become larger (due to the scaling factor $t$ inside the exponential) and $p_t$ now becomes more uni-modal, picking the closest mode (as there is little time left to go to the other modes, making them more unlikely). Thus, our task of finding a subgroup within a larger group is challenging precisely because the modes are "symmetric" (by definition of the subgroup).

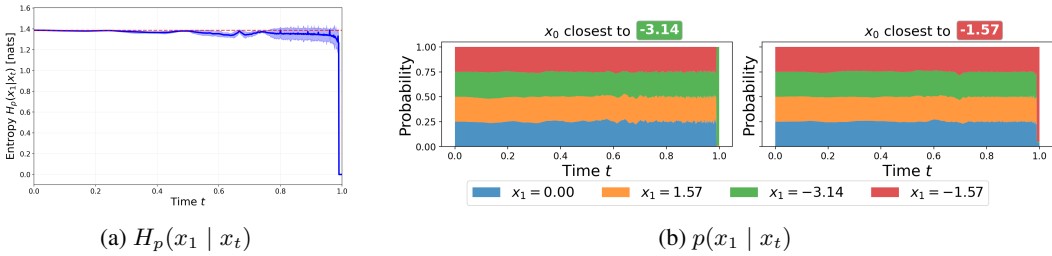

(a) $H_p(x_1 \mid x_t)$

(b) $p(x_1 \mid x_t)$

Figure 7: Left: The entropy of the posterior $p(x_1 \mid x_t)$ is generally uniform until $t \approx 1$.
Right: $x_0$ tends to converge to the closest mode/component (e.g., $x_0$ closest to $-\pi$ tends to converge to $-\pi$), but the posterior remains nearly uniform until $t \approx 1$.

**Flow Matching Directly on Group Elements.** To verify our hypothesis, we perform flow matching directly on the group elements from $\mathrm{SO}(2)$ to $C_4$. The group elements can be parameterized by a scalar $\theta$, allowing us to analyze the behavior of the learned vector field more easily. Each data sample $x_1$ is a scalar from the set $\{-\pi, -\pi/2, 0, \pi/2\}$ and the source distribution is $q = \mathcal{U}[-\pi, \pi]$. We use the same network architecture as in the 2D experiments. We can further compute the reverse conditional or the posterior $p_t(x_1 \mid x_t)$ that can help verify two things: 1) that $x_t$ remain nearly stationary until $t$ approaches $1$, i.e. $p(x_l \mid x_t)$ remains nearly uniform, and 2) $x_t$ chooses the closest group element $x_1$. The exact details of how $p_t(x_1 \mid x_t)$ is computed is given in Appendix D.

Figure 29 in Appendix C.6 visualizes the progression of $x_t$ over time for $100$ samples, showing that $C_4$ symmetry is clearly identified. Figure 7a plots the entropy of the posterior, and shows that even in this simple scenario, the entropy remains near the maximum (uniform distribution) until $t$ approaches $1$. Figure 7b visualizes the posterior $p(x_1 \mid x_t)$ averaged over $1{,}000$ samples and separated into $4$ classes, depending on the original $x_0$ and its nearest mode in $\{-\pi, -\pi/2, 0, \pi/2\}$ (only two shown, other classes are similar, see Figure 30b in Appendix C.6). For all $4$ classes, we observe that $p(x_1 \mid x_t)$ remains nearly uniform until around $t = 0.9$, when it becomes uni-modal to the closest mode/component, e.g., for all $x_0$ that were closest to the mode $\mu = -\pi$, then the posterior $p(x_1 \mid x_t)$ converges to $\mu$. This shows both that the learned vector field remains nearly $0$ for most timesteps, only converges when little time is left, and converges to the closest group element.

Note that this differs from the standard FM setting, where the flow is defined on a high-dimensional space. Bertrand et al. (2025) find that for high-dimensional flow matching with images, the entropy of the posterior drops at small times, transitioning from a stochastic phase that enables generalization to a non-stochastic phase that matches the target modes. In our low-dimensional setting of symmetry discovery, we find the opposite occurs: the entropy stays near uniform until large $t$, where it suddenly converges. We term this phenomenon "last-minute mode convergence".

Figure 8 shows the evolution of a uniform grid over $\mathrm{SO}(2)$ over time. As we hypothesized, we can see from the velocity arrows that particles move slowly closer to the midpoint between two modes ($t = 0$ to $t = 0.60$) but the velocities are relatively small. From $t = 0.70$ onwards, we can see

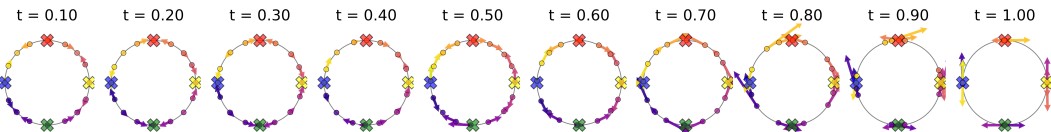

Figure 8: SO(2) to $C_4$ group elements: Evolution of $x_0$'s and their velocities over time.

that the shift towards the closest modes and the velocities become increasingly large until $t = 1$. Figure 31 (Appendix C.6) shows a clearer picture of how the velocity evolves over time. From $t = 0$ to $t = 0.6$, we see that the $x_t$'s are nudged towards the midpoints between modes. Interestingly, we see some oscillating behavior where the vector field flips sign, suggesting that the net velocity hovers close to 0 with no defined pattern. From $t = 0.8$ onwards, we see that points $x_t$ are pushed to the closest $x_1$ mode, with increasing velocities. With near zero net velocity until $t = 0.6$, these plots also support the observation of "last-minute mode convergence".

## 6 DISCUSSION

Despite the challenges revealed in our experiments, this work demonstrates that generative models, particularly flow matching, offer a promising avenue for symmetry discovery. We introduced a novel approach to symmetry discovery using flow matching over Lie groups, where we learn a map from a larger prior group to the data symmetries. Our experiments revealed both successes and fundamental challenges. In the 2D case, we successfully discovered discrete symmetries from several different continuous prior groups, including the dihedral group by learning a flow over the complex domain. The method naturally discovers the closest group element rather than exact transformations, which may be advantageous for handling approximate symmetries in real data. However, our 3D experiments exposed some limitations: while our method can discover smaller subgroups such as the tetrahedral group, it fails for higher order groups such as the octahedral group. Our analysis uncovered a "last-minute convergence" phenomenon, that the symmetric arrangement of the target modes causes the learned vector field to remain near zero for most of the trajectory, only converging to the nearest mode as $t$ approaches 1. Our simplified flow matching experiment supported this claim and we showed that the posterior entropy remains near uniform for most timesteps. We introduced a power time schedule to skew the sampling towards timesteps near 1. Our heuristic allows use to discover the more complex octahedral group, but still fails for higher-order groups.

### 6.1 LIMITATIONS AND FUTURE WORK

There are several limitations to our work. While we demonstrate feasibility on simple and noise-free point clouds, scaling to noisy real-world datasets remains an open question. Our method is currently limited to discovering symmetries within the parameterization of the hypothesis group, and it is still an open question as to whether `LieFlow` will scale to real-world datasets. Our method requires fitting a distribution over the prior group, which may impact the range of symmetries it can discover depending on the parameterization and boundaries. Although we have analyzed the cause behind last-minute mode convergence and proposed a time schedule according to the power distribution, it is still unclear how to discover complex symmetries. An important future direction is to investigate Riemannian diffusion–based formulations, which can sidestep the issue of discontinuities of the target vector field at the cut locus (Huang et al., 2022; Zhu et al.; Mangoubi et al.), and may alleviate the last-minute mode convergence behavior observed in LieFlow. Another important future direction is to improve numerical stability and scalability on high-dimensional or non-compact groups, enabling LieFlow to fully leverage its group-agnostic framework beyond the lower-dimensional settings demonstrated here.

Future directions include developing specialized architectures with group-aware inductive biases, incorporating additional supervision signals to guide convergence, and extending the method to handle approximate symmetries common in real-world data. The integration of discovered symmetries into downstream equivariant networks also remains an important open question. Despite current limitations, this work demonstrates that flow matching offers a promising path toward automatic symmetry discovery, lending us insights into the data and offering improved sample efficiency and generalization for downstream tasks.

## LLM USAGE

LLMs were used to revise sentences, correct grammar, and draft portions of the introduction, method, and discussion (Sections 1, 4 and 6). They were also used to generate parts of the model code and most of the visualization code.

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

## A DERIVATION OF THE TARGET FLOW FIELD

This section provides a detailed derivation of the target vector field $u_t(x_t \mid x_1)$ used in our Lie group flow matching framework. We begin by introducing flow matching on Lie groups, followed by the derivation of the target vector field on the data space induced by the group action.

**Interpolants on Lie Groups.** Denote the left action of $G$ on itself for any $g \in G$ as $L_g : G \to G, \hat{g} \mapsto g\hat{g}$. Let $T_g G$ be the tangent space at $g \in G$. The push-forward of the left action is $(L_g)_* : T_{\hat{g}}G \to T_{g\hat{g}}G$, which transforms tangent vectors at $T_{\hat{g}}G$ to tangent vectors at $g\hat{g} \in G$ via the left action.

To derive FM on a Lie group, we need to define the interpolant as in the Euclidean case. An exponential curve starting at $g_0, g_1 \in G$ can be defined by (Sherry & Smets, 2025):

$$\gamma : [0, 1] \to G; \quad t \mapsto g_0 \exp(t \log(g_0^{-1}g_1)). \tag{6}$$

This curve starts at $\gamma(0) = g_0$ and ends at $\gamma(1) = g_1$. The multiplicative difference is $g_0^{-1}g_1$ and the logarithm maps it to the Lie algebra, giving the direction from $g_0$ to $g_1$. The scaled version is then exponentiated, recovering intermediate group elements along this curve. Using these exponential curves, we can define the target vector field as

$$u_t(g_t \mid g_1) = \frac{(L_{g_t})_* \log(g_t^{-1}g_1)}{1 - t}. \tag{7}$$

**Defining the Target Vector Field.** Note that we couple source and target by an explicitly sampled group action. According to equation 4, given $x_1 \sim q$ and $g \sim p(G)$, and set $x_0 = gx_1$ and $A := \log(g^{-1}) \in \mathfrak{g}$. Consider the group curve $g_t = g\exp(tA)$ so that $g_t^{-1} = \exp((1-t)A)$ and the induced path $x_t = g_t x_1 = x_0 \exp(tA)$. Plugging $g_1 = e$ into the group interpolant equation 7, we have:

$$u_t(g_t \mid e) = \frac{(L_{g_t}) * \log(g_t^{-1})}{1 - t} = (L_{g_t}) * A, \tag{8}$$

which is a time independent target field in the Lie algebra along this curve. To obtain the conditional target on the data path, we first define the orbit map $\phi_{x_1} : G \to \mathcal{X}$, $\phi_{x_1}(g) = gx_1$. Its differential at $g_t$ is given as $(d\phi_{x_1})_{g_t} : T_{g_t}G \longrightarrow T_{x_t}\mathcal{X}$, where $x_t = \phi_{x_1}(g_t)$. Note that $(L_{g_t})* : \mathfrak{g} \to T_{g_t}G$. Compose them together, we obtain the pushforward operator $J_{x_t} := (d\phi_{x_1})_{g_t} \circ (L_{g_t})* : \mathfrak{g} \longrightarrow T_{x_t}\mathcal{X}$. Then, we push the group target to the data path and obtain the conditional target on data:

$$u_t(x_t \mid x_1) = (d\phi_{x_1})_{g_t}\big[u_t(g_t \mid e)\big] = (d\phi_{x_1})_{g_t}\big[(L_{g_t}) * A\big] = J_{x_t}(A) \in T_{x_t}\mathcal{X}. \tag{9}$$

We define the stabilizer at $x_t$ as $\mathrm{Stab}(x_t) = \{s \in G : s \cdot x_t = x_t\}$ and $\mathfrak{s}_{x_t}$ is the Lie algebra of the stabilizer. According to the Orbit-stabilizer theorem, we have:

$$\mathrm{Ker}J_{x_t} = \mathfrak{s}_{x_t}, \qquad \mathrm{Im}\, J_{x_t} = T_{x_t}\big(G \cdot x_1\big).$$

This means that if the stabilizer of the data is trivial, the pushforward operator is bijective. Consequently, every tangent vector $u_t(x_t|x_1) \in T_{x_t}(G \cdot x_1)$ has a unique Lie-algebra coordinate $A \in \mathfrak{g}$ such that $u_t(x_t \mid x_1) = J_{x_t}(A)$. Using this coordinate identification, for simplicity, we can write $u_t(x_t|x_1) \equiv A$ as the target vector field.

# B    TRAINING DETAILS

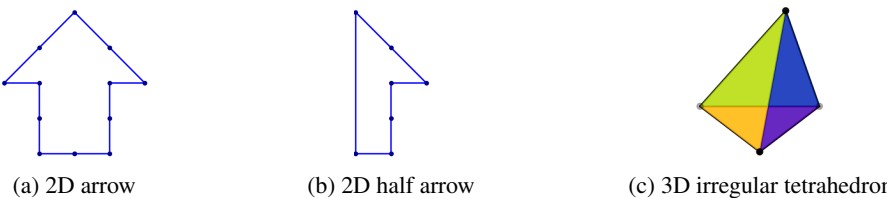

|          |               |                            |
|:--------:|:-------------:|:--------------------------:|
| (a) 2D arrow | (b) 2D half arrow | (c) 3D irregular tetrahedron |

Figure 9: Canonical objects used for generating the 2D (a,b) and 3D (c) datasets

**2D Datasets**  For the 2D point cloud datasets, we use canonical arrow and half-arrow objects as shown in Figure 9a,b. We generate datasets containing discrete symmetries, $C_4$ and $D_4$ transforming the canonical objects. The prior known groups are $\mathrm{SO}(2), \mathrm{GL}(2,\mathbb{R})^+$ for discovering $C_4$ and $\mathrm{GL}(2,\mathbb{C})$ for discovering $D_4$. For $\mathrm{SO}(2)$, we use a uniform distribution over the parameter $\theta \sim \mathcal{U}[-\pi, \pi]$ and generate the $2 \times 2$ rotation matrices. For the $\mathrm{GL}(2)$ groups, we fit a pushforward distribution via the exponential map from a uniform distribution $\mathcal{U}[-\pi/2, \pi/2]$ over the Lie algebra coefficients and use the standard basis for the generators. On the network side, we use a small MLP and concatenate time $t$ to $x_t$ as inputs.

**3D Datasets**  We use an irregular tetrahedron with no self-symmetries as shown in Figure 9c and generate datasets containing $\mathrm{Tet}, \mathrm{Ico}$ symmetries. We consider the prior group $\mathrm{SO}(3)$ and use a uniform distribution over it by using Gaussian normalization over unit quaternions and transforming them into $3 \times 3$ matrices.

For the 3D case, we use a similar but wider network architecture as the 2D datasets and use a sinusoidal time embedding.

## C  RESULTS

This section contains additional results and visualizations for both 2D and 3D datasets.

### C.1  2D DATASETS

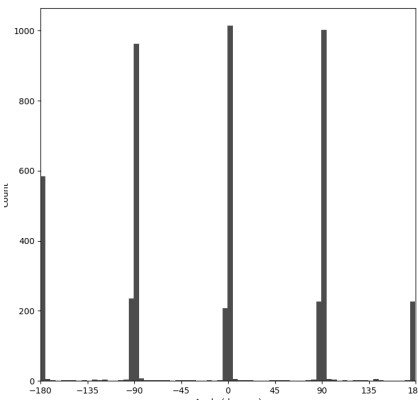

Figure 10: Histogram of 5000 angles from the generated transformation matrices for $\text{SO}(2)$ to $C_4$.

Figure 10 is the histogram of the angles of the generated transformation matrices for $\text{SO}(2)$ to $C_4$. We can see that the model learns to generate angles close to multiples of $\pi/2$, corresponding to $C_4$ elements.

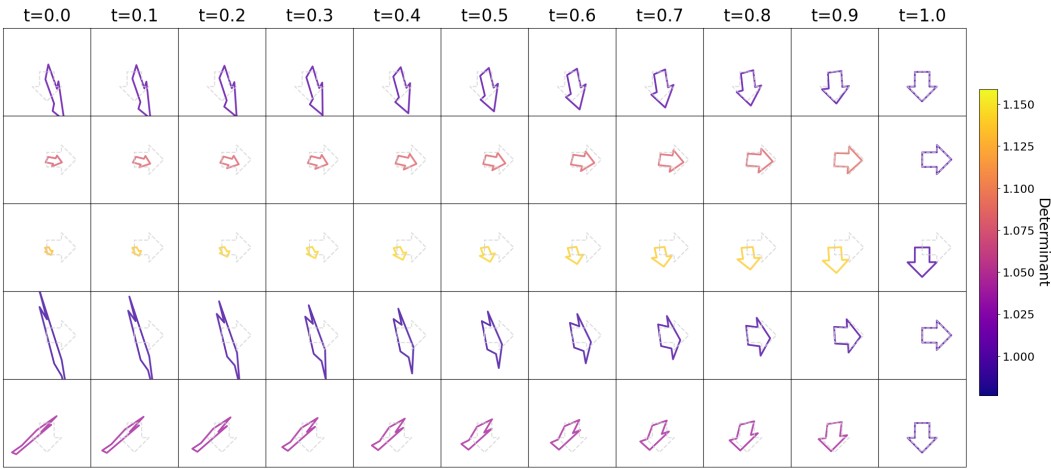

Figure 11: $\text{GL}(2)^+$ to $C_4$: Time progression of $x_t$ when generating 5 samples over 20 steps. The gray arrow shows the original $x_1$ and the color represents the determinant of the generated transformation matrix.

Figure 11 shows the time progression of $x_t$ when generating 5 samples over 20 steps for $\text{GL}(2)^+$ to $C_4$. The gray arrow shows the original $x_1$ and the color represents the determinant of the generated transformation matrix. We can see that the transformed point clouds converge to the closest group element in the orbit.

## C.2 3D DATASETS: WITH TIME SAMPLED FROM UNIFORM DISTRIBUTION

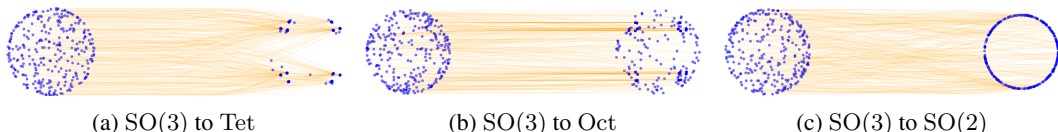

(a) SO(3) to Tet       (b) SO(3) to Oct       (c) SO(3) to SO(2)

Figure 12: 3D Datasets: A 2D PCA visualization of trajectories from $t = 0$ (left) to $t = 1$ (right) of the centroids of the transformed objects over 100 samples.

Figure 12 visualizes the trajectories of the centroids over time for 100 samples, and PCA is performed to project them onto the 2D plane. We can see that our model seems to learn Tet and SO(2) elements, creating some visible clusters, but fails to discover Oct symmetries.

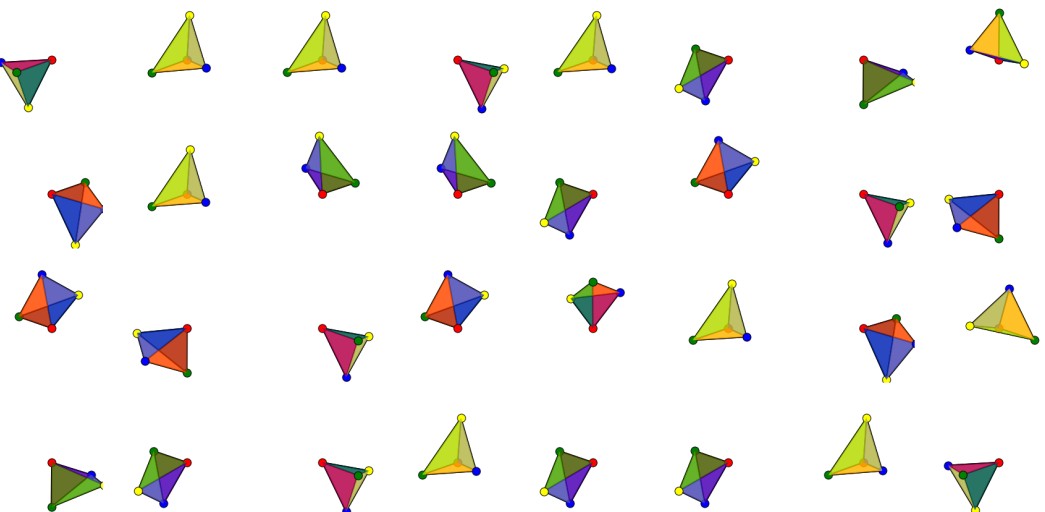

Figure 13: SO(3) to Tet: generated transformed point clouds from random $x_1$ in the test set.

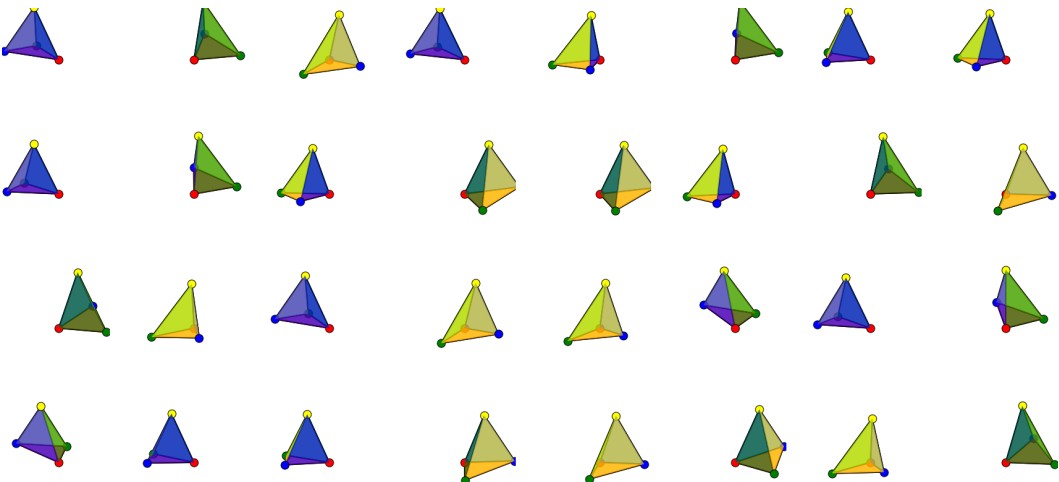

Figure 14: SO(3) to SO(2): generated transformed point clouds from random $x_1$ in the test set.

Figure 13, Figure 14 and Figure 15 show generated transformed point clouds from random $x_1$ in the test set for Tet, SO(2) and Oct respectively. We can see several repeated point clouds, and see that some samples are rotations or reflections of other samples.

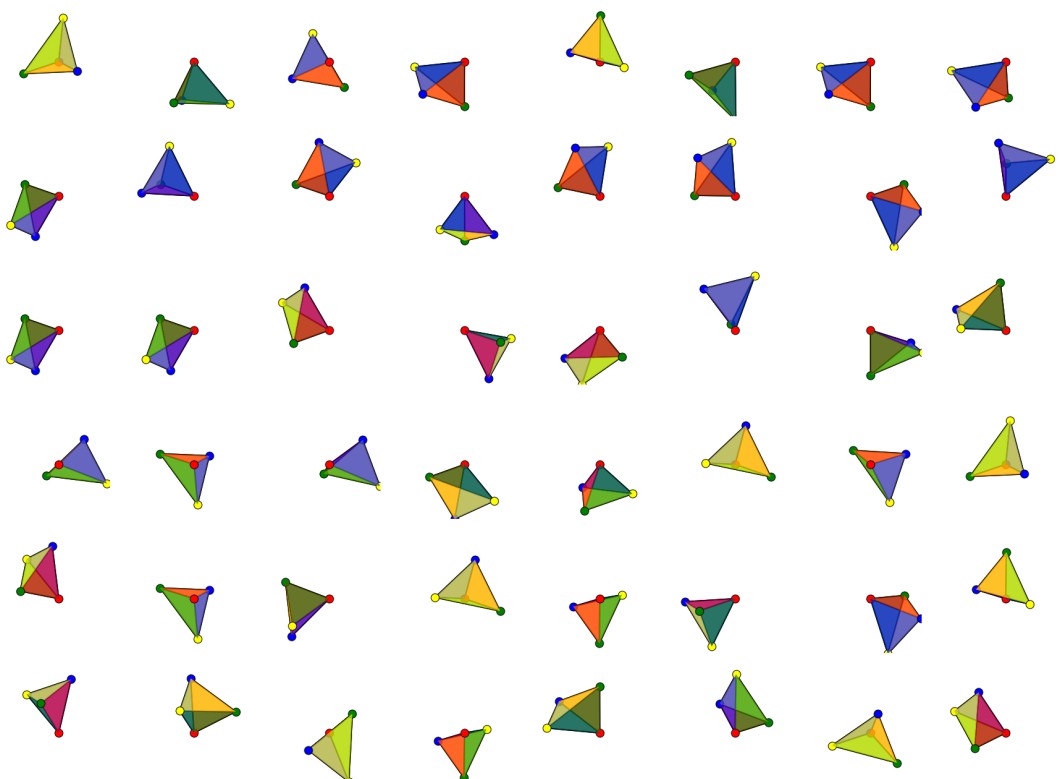

Figure 15: SO(3) to Oct: generated transformed point clouds from random $x_1$ in the test set.

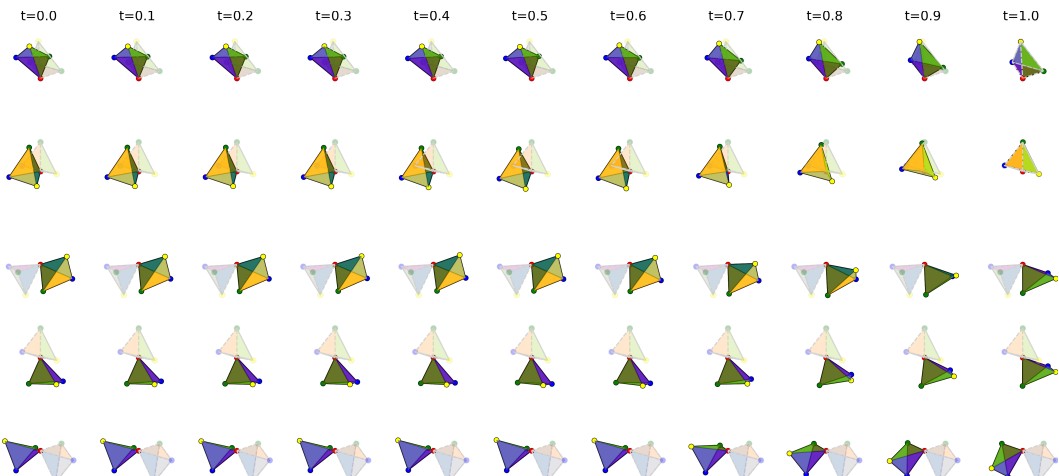

Figure 16: SO(3) to Tet: time progression of $x_t$ when generating 5 samples over 100 steps. The transparent tetrahedron indicates the original $x_1$.

We also show the intermediate $x_t$ during generation in Figure 16, Figure 17 and Figure 18. For the Tet, we see that the transformed point clouds converge to the closest group element in the orbit. For the SO(2) group, we can see that the model produces rotations to turn one node of the tetrahedron to be aligned with the $z$-axis and one triangle face to be in the $xy$-plane, which corresponding to SO(2) symmetries around the $z$-axis. For the Oct group, we can see that the FM model produces outputs close to the identity at most steps and fails to properly discover Oct.

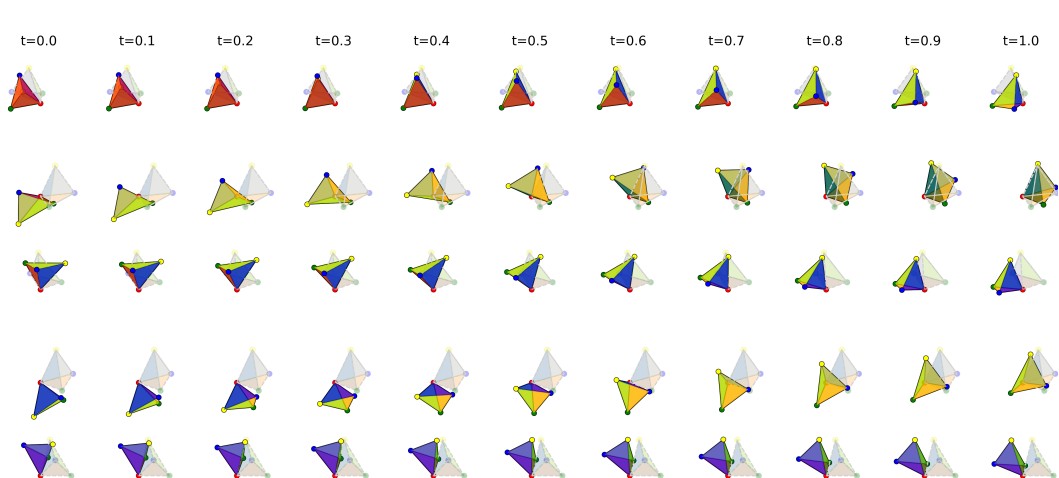

Figure 17: SO(3) to SO(2): time progression of $x_t$ when generating 5 samples over 100 steps. The transparent tetrahedron indicates the original $x_1$.

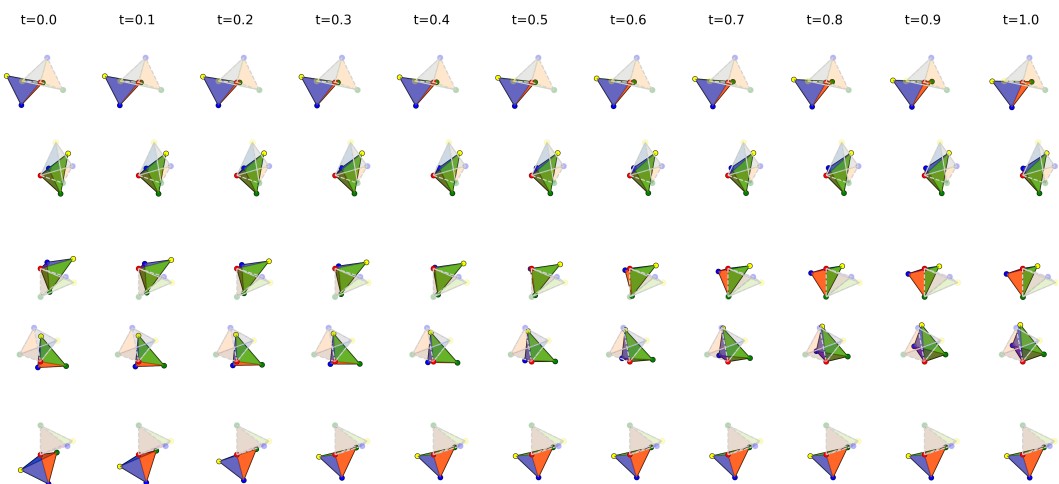

Figure 18: SO(3) to Oct: time progression of $x_t$ when generating 5 samples over 100 steps. The transparent tetrahedron indicates the original $x_1$.

## C.3    3D DATASETS: WITH TIME SAMPLED FROM POWER DISTRIBUTION

In this section, we show additional results for the 3D datasets with time sampled from the power distribution. The power distribution density is shown in Figure 19.

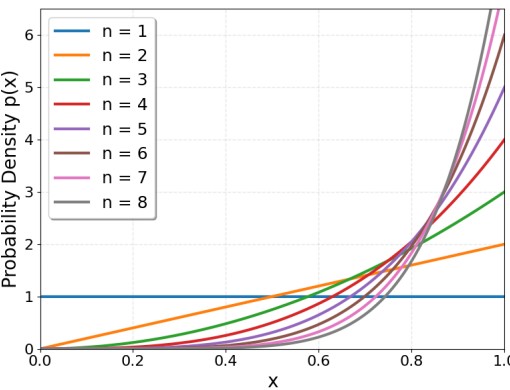

Figure 19: Power distribution density

Figure 19 shows the density of the power distribution for different skewness values.

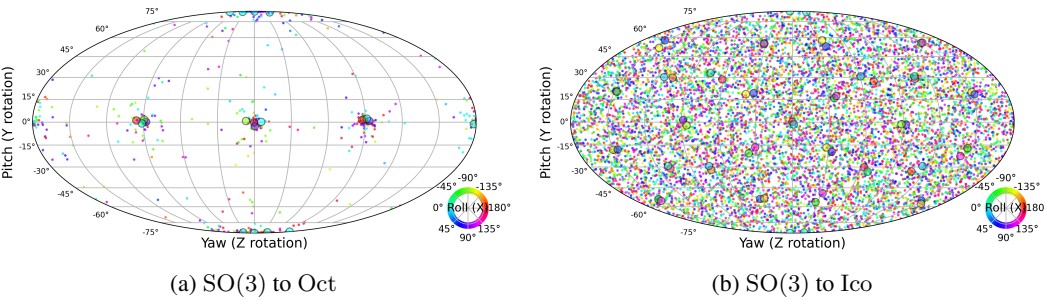

(a) SO(3) to Oct                                        (b) SO(3) to Ico

Figure 20: 3D datasets with power time schedule: visualization of 5,000 generated elements of SO(3) by converting them to Euler angles. The first two angles are represented spatially on the sphere using Mollweide projection and the color represents the third angle. The elements are canonicalized by the original random transformation and the ground truth elements of the target group are shown in circles with black borders. The points are jittered with uniformly random noise to prevent overlapping.

Figure 20 shows the generated transformations when using the power distribution time schedule. We can see that flow matching now works for octahedral group, but still fails to discover the icosahedral group.

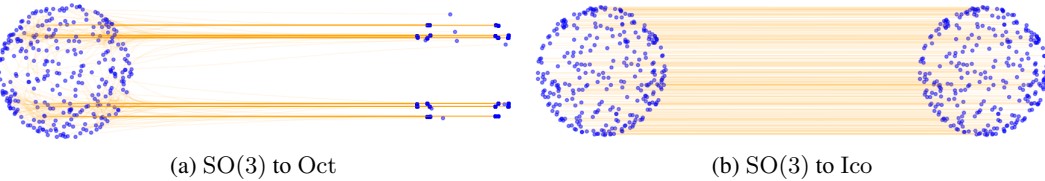

(a) SO(3) to Oct                                        (b) SO(3) to Ico

Figure 21: 3D datasets with power time schedule: A 2D PCA visualization of trajectories from $t = 0$ (left) to $t = 1$ (right) of the centroids of the transformed objects over 100 samples.

Figure 21 visualizes the centroid trajectories after performing PCA. The octahedral case shows clear clusters, while the Ico case seems to output only identity transformations.

## C.4 DISCOVERING SYMMETRY IN MULTI-OBJECT DATASET

In this section, we show additional results for discovering symmetries in a multi-object dataset. We create a dataset consist of four different irregular 3D objects (Figure 22): tetrahedron, triangular prism, cube, octahedron. We use Tet symmetries as the target group and $SO(3)$ as the prior group. The training details are the same as in the single-object 3D experiments.

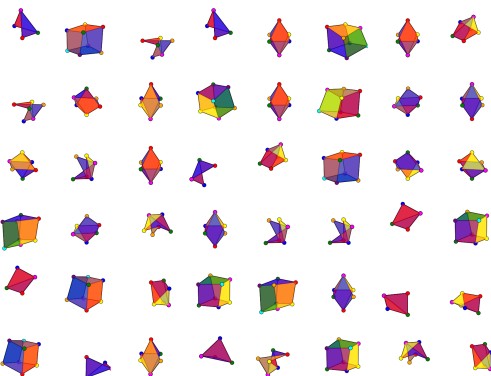

Figure 22: Four different irregular 3D objects used in the multi-object dataset

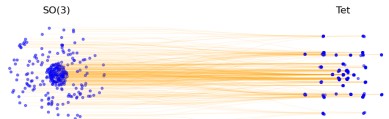

Figure 23: A 2D PCA visualization of trajectories from $t = 0$ (left) to $t = 1$ (right) of the centroids of the transformed objects over 100 samples.

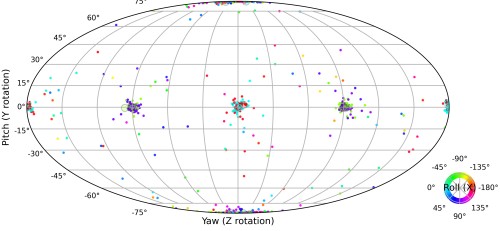

Figure 24: Visualization of 5,000 generated elements of $SO(3)$ by converting them to Euler angles. Same as Figure 20.

Figure 23 visualizes the centroid trajectories after performing PCA. We can see that the model learns to create clusters corresponding to Tet symmetries. Figure 24 shows the generated transformations, and we can see that the model discovers the Tet group elements. Figure 25 shows the time progression of $x_t$ when generating 5 samples over 100 steps, and we can see that the transformed point clouds converge to the closest group element in the orbit. Finally, to quantify the symmetry discovery performance, we compute the Wasserstein-1 distance between the discovered group elements and the ground-truth Tet elements, obtaining a distance of 0.10, while for LieGAN Yang et al. (2023) baseline we obtain a distance of 2.20(1 Lie generators) 1.95(3 Lie generators), and 1.58(6

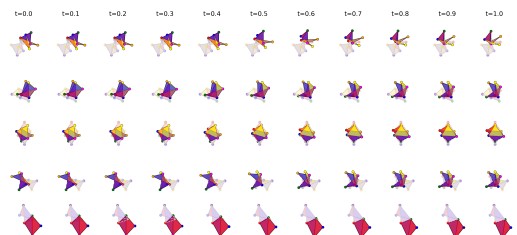

Figure 25: Time progression of $x_t$ when generating 5 samples over 100 steps. The transparent object indicates the original $x_1$.

Lie generators), which shows that our method significantly outperforms the baseline in discovering the correct symmetries.

### C.5 DISCOVERING STRUCTURED-PRESERVING TRANSFORMATION

To demonstrate that our method can discover structure-preserving transformations, we create a dataset by applying $z$ axis rotations from $\mathrm{SO}(2)$ with angles sampled from a Gaussian distribution $\mathcal{N}(0, \pi/4)$ to an irregular tetrahedron. Thus, the data distribution is not uniform along the $\mathrm{SO}(2)$ orbits, but the underlying symmetry structure is still $\mathrm{SO}(2)$. We use $\mathrm{SO}(3)$ as the prior group and train our LieFlow model as before.

The results are shown in Figure 26, Figure 27, and Figure 28. Figure 26 visualizes the centroid trajectories after performing PCA. We can see that the model learns to create an incomplete circular pattern, corresponding to leanring a distribution over $\mathrm{SO}(2)$ group elements. Figure 27 shows the generated transformations, and we can see that the elements are concentrated along a one-dimensional manifold, and the samples in the center are more dense than those in the periphery, corresponding to learning a distribution over $\mathrm{SO}(2)$ group elements. Figure 28 shows the histogram of the angles rotated around the $z$-axis extracted from the generated transformations. We can see that the distribution of angles closely matches Gaussian distribution. The Wasserstein-1 distance between the discovered distribution and the ground-truth Gaussian distribution is $0.282$, showing that our method can accurately recover the underlying structure-preserving transformations.

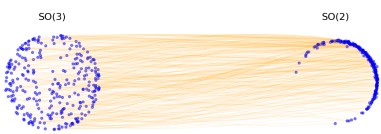

Figure 26: A 2D PCA visualization of trajectories from $t = 0$ (left) to $t = 1$ (right) of the centroids of the transformed objects over 100 samples.

### C.6 FLOW MATCHING ON GROUP ELEMENTS

This section contains additional visualizations for the flow matching experiment on group elements from $\mathrm{SO}(2)$ to $C_4$ as described in Section 5.4.

Figure 29 shows the time progression of $x_t$ when generating 100 samples from $t = 0$ to $t = 1$. We can see that the samples converge to the closest group element in the orbit.

Figure 30b in Figure 30 shows the posterior $p(x_1 \mid x_t)$ for all 4 classes, depending on the original $x_0$ and its nearest mode in $\{-\pi, -\pi/2, 0, \pi/2\}$. We can see that for all 4 classes, the posterior $p(x_1 \mid x_t)$ remains nearly uniform when $t$ closes to 1.

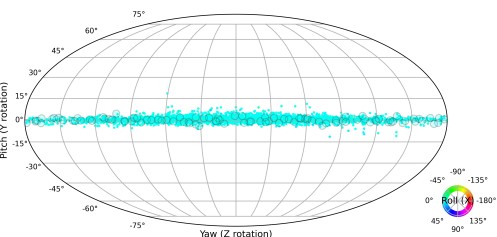

Figure 27: Visualization of 5,000 generated elements of SO(3) by converting them to Euler angles. Same as Figure 20.

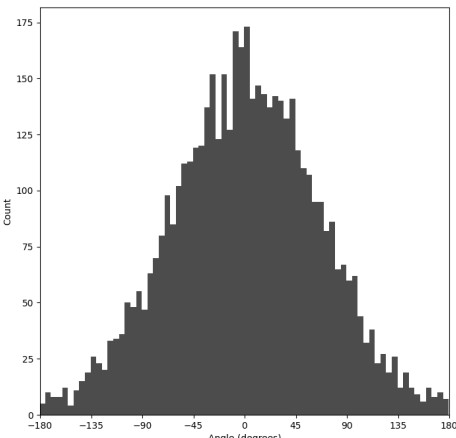

Figure 28: Histogram of 5,000 angles rotated around the $z$-axis extracted from the generated transformations.

Figure 30 visualizes the posterior $p(x_1 \mid x_t)$, divided into 4 buckets depending on the initial position $x_0$. We see that the posterior remains nearly uniform until $t = 0.95$, and then $x_t$ converges to the mode that was the closest when $t = 0$.

Figure 31 visualizes how the velocities change over time. Until $t = 0.6$, the velocities are close to 0 but slightly push the $x_t$'s towards the midpoints between modes. From $t = 0.8$ onwards, we see that points $x_t$ are pushed to the closest $x_1$ mode, with increasing velocities.

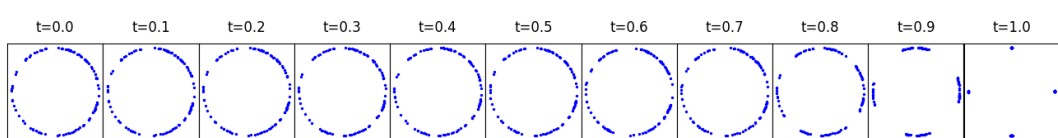

Figure 29: SO(2) to $C_4$ group elements: Visualization of 100 samples over time.

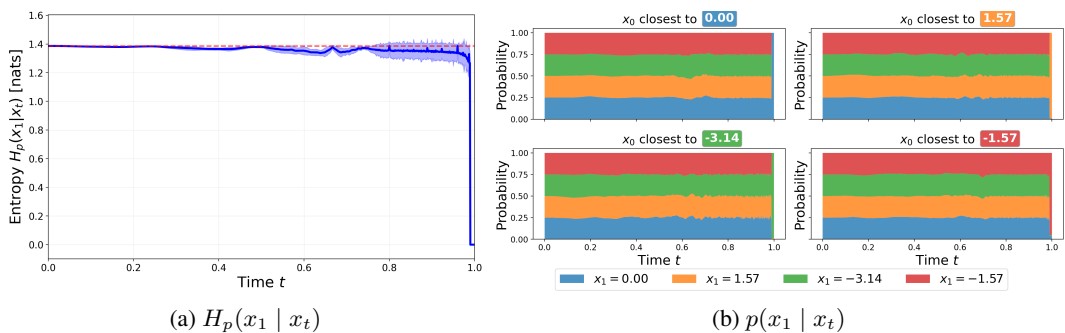

(a) $H_p(x_1 \mid x_t)$

(b) $p(x_1 \mid x_t)$

Figure 30: Visualizations of the posterior $p(x_1 \mid x_t)$. The left graph shows the entropy of the posterior is generally uniform until $t \approx 1$. The right graph visualizes $p(x_1 \mid x_t)$ for each target mode and demonstrates that the generated samples, depending on the position of $x_0$, converge to the closest mode.

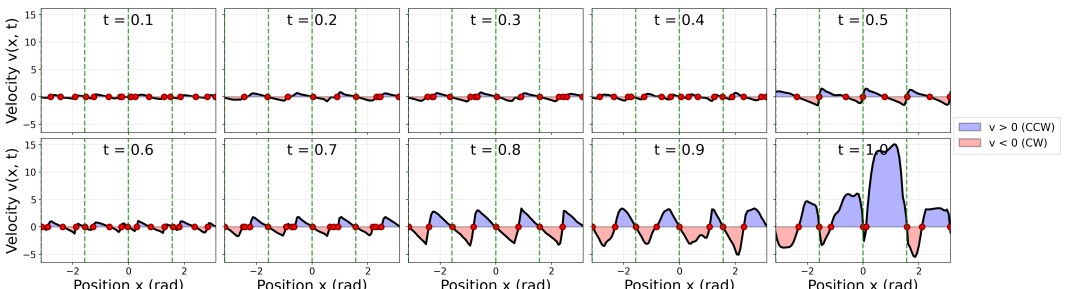

Figure 31: SO(2) to $C_4$ group elements: velocity phase portraits over time.

# D    FLOW MATCHING DIRECTLY ON GROUP ELEMENTS

**Computation of the posterior** $p(x_1 \mid x_t)$    To compute the posterior, we use Bayes' rule as

$$p(x_1|x_t) = \frac{p(x_t|x_1)p(x_1)}{p_t(x_t)}, \tag{10}$$

where $p(x_t \mid x_1)$ is the likelihood and $p(x_1)$ is the data distribution. Since flow matching learns a velocity field rather than explicit conditional distributions, we cannot directly compute the likelihood and need to invert the flow and integrate the continuity equation.

We first sample initial points $x_0$ from a uniform prior over $[-\pi, pi)$ and forward simulate using the learned velocity field $v_\theta$ to obtain the trajectories $x_t$ and their log probabilities through the continuity equation. To evaluate the likelihood $p(x_t \mid x_1)$ for each candidate $x_1$, we invert the linear interpolation $x_t = (1-t)x_0 + tx_1$ to find $x_0 = (x_t - tx_1)/(1-t)$, then integrate the divergence along the trajectory from this inverted $x_0$. Special handling is required at the boundaries: at $t = 0$, the posterior equals the prior $p(x_1)$ (which we know in this simple scenario) due to independence, while near $t = 1$, we use a Gaussian approximation to avoid numerical instability from division by $(1-t)$.

---

**Algorithm 4** Compute Posterior $p(x_1 \mid x_t)$ for Flow Matching

---

**Require:** Velocity field $v_\theta(x,t)$, prior $p_0 \sim \mathcal{U}[-\pi, \pi]$, target $p_1(x_1) = \sum_{k=1}^{K} \frac{1}{K}\delta(x_1 - \mu^{(k)})$, time
    steps $T$
1: **Initialize:** Sample $\{x_0\}^N \sim p_0$
2:
3: **Forward Simulation:**
4: **for** $t \in \{0, \frac{1}{T}, \dots, 1\}$ **do**
5:    $x_t \leftarrow \text{ODESolve}(v_\theta, x_0, [0, t])$             ▷ Integrate velocity field
6:    $\log p_t(x_t) \leftarrow \log p_0(x_0) - \int_0^t \nabla \cdot v_\theta(x_s, s)ds$    ▷ Continuity equation
7: **end for**
8:
9: **Posterior Computation:**
10: **for** $t \in \{0, \frac{1}{T}, \dots, 1\}$ **do**
11:    **if** $t = 0$ **then**
12:        $p(x_1 \mid x_0) \leftarrow p_1(x_1)$              ▷ Independent at $t = 0$
13:    **else if** $t \approx 1$ **then**
14:        **for** $k = 1$ to $K$ **do**
15:            $\log p(x_t \mid x_1^{(k)}) \leftarrow -\frac{\|x_t - x_1^{(k)}\|^2}{2\sigma^2}$, where $\sigma = 0.1$    ▷ Gaussian Approximation
16:        **end for**
17:    **else**
18:        **for** $k = 1$ to $K$ **do**
19:            $x_0^{(k)} \leftarrow \frac{x_t - t \cdot x_1^{(k)}}{1-t}$              ▷ Invert linear interpolation
20:            Forward simulate path from $x_0^{(k)}$ to time $t$ using $v_\theta$ and continuity equation
21:            $\log p(x_t \mid x_1^{(k)}) \leftarrow \log p_0(x_0^{(k)}) - \int_0^t \nabla \cdot v_\theta(x_s, s)ds$
22:        **end for**
23:    **end if**
24:    **for** $k = 1$ to $K$ **do**
25:        $\log p(x_1^{(k)} \mid x_t) \leftarrow \log p(x_t \mid x_1^{(k)}) + \log p_1(x_1^{(k)})$    ▷ Bayes' Rule
26:    **end for**
27:
28:    $p(x_1^{(k)} \mid x_t) \leftarrow \frac{\exp(\log p(x_1^{(k)}|x_t))}{\sum_k \exp(\log p(x_1^{(k)}|x_t))}$    ▷ Normalize over $K$ components
29: **end for**
30: **return** Posterior $\{p(x_1 \mid x_t)\}$ for all time steps

---

