# OpenReview forum: "Discovering Lie Groups with Flow Matching"
_ICLR.cc/2026/Conference — Submitted to ICLR 2026_

### Official Review · Reviewer_cRXJ · 2025-10-30

**Soundness:** 2
**Presentation:** 2
**Contribution:** 3
**Rating:** 2
**Confidence:** 3

**Summary:**

This paper seeks to identify the subgroup which preserves a distribution of data, given a matrix group acting on the data space. This task is accomplished by means of flow matching on Lie groups. Given a parametric family of group elements, a prior distribution is assumed on the group, and the flow matching task maps the prior distribution to a new distribution, which new distribution is representative of the subgroup of distribution-preserving group elements (under the group action).

**Strengths:**

1. This appears to be a novel approach for symmetry detection and it may be well-suited for discovering subgroups that preserve data distributions.
2. This work demonstrates significant potential, though the weaknesses must be addressed. But this seems like a promising research direction that I would like to see developed.

**Weaknesses:**

1. *Limited experimental comparison.* I see no experimental comparison with existing methods of symmetry detection. Such comparison is crucial so that readers can understand whether the novel method performs better than existing methods at similar tasks. As scalability is beginning to be a topic of interest in symmetry detection, it seems that a comparison of running times is also appropriate.

2. *Experimental results do not validate some of the authors' claims.* Around line 175, it is said that $SO(d)$ and $GL(d)$ are certainly feasible. But there is no theoretical or empirical justification for this statement. In fact, the notion that the method struggles when the symmetry group increases in size seems to cast doubt on the ability of the method to generalize well to higher dimensions, where the subgroups will consist of many more elements. Additionally, the authors claim that their method is applicable to continuous symmetries, as one only need look for an output distribution which is "spread." This seems to open up a can of worms relating to approximate symmetries, which could potentially be addressed with a simple 3d experiment--perhaps when the point cloud takes a familiar shape, such as a paraboloid or (even better due to mixed symmetry types) a hyperboloid of two sheets.

3. *The writing is unclear at times.* Though it appears that there are typos throughout, there are potentially more serious writing issues that disable proper ingestion of the methodology. (some of these moments may arise from a misunderstanding on my part.) For example, in the *training* portion of 4.2, it is not clear what is meant by the statement "$x_0$ is now a sample from the prior and lives in the group $G$:" how can this be, given that $x_0$ is defined as $g x_1$, being a transformation away from $x_1$ which was sampled from the data distribution? Next, the authors claim to predict $A$ given $x_t$: and yet, in algorithm 1, $A$ is directly calculated from $g$, and it appears that $v_{\theta}(x_t,t)$ is being predicted. Algorithm 2 may also be missing a line.

4. *The topic of flows and vector fields is not entirely precise.* A one-parameter flow arises from a time-*independent* vector field, not a time-*dependent* vector field. The time-*dependent* case introduces an additional parameter *beyond the flow parameter*, so that the "flow" for a time-*dependent* vector field should actually have two parameters attributed to it. As constructed, the map $\phi_t(x)$, likely gotten by evaluating the flow parameter at $0$, is actually not a flow and is better described as an evolution operator.

5. *Missing some related work.* There are additional works that should be considered, owing in particular to this work's alleged tie between flows and symmetry: "Symmetry Discovery Beyond Affine Transformations" (Shaw et al., 2024), "Learning Infinitesimal Generators of Continuous Symmetries from Data" (Ko et al., 2024), and "A Unified Framework to Enforce, Discover, and Promote Symmetry in Machine Learning" (Otto et al., 2023).

6. *Sub-optimal presentation.* The figures are hard to read (font size). There appear to be typos and other oddities: lines 59-61 may be missing content, line 81 should have "at the identity" instead of "to the identity", typo around line 161, etc.

**Questions:**

1. I don't quite understand how the distribution-preserving group elements $h$ are distinguished from the others. This may be related to weakness 3, where I felt the writing was unclear around the discussion of the algorithms.

2. There has been some level of discussion in the community about not merely distribution-preserving transformations, but also structure-preserving transformations. That is, group actions that preserve the manifold structure of the data, not necessarily the way in which the data is distributed on the manifold. Would this method be adaptable to the manifold-preserving case?

3. My other questions relate to the weaknesses noted above: (a) time to compute symmetries; (b) experiments involving continuous symmetry; (c) comparison with other methods.

---

> ### Author Response · Authors · 2025-12-03
> **Author Response [1/2]**
>
> We thank the reviewer for their detailed and constructive analysis and address their concerns below.
>
> 1. **Baseline comparison**
>
> In the revision, we include results using LieGAN [1], which is the closest prior work that also learns to discover group actions. We thank the reviewer for pointing out the need for comparison with existing symmetry discovery methods. As an additional new contribution, we introduce the Wasserstein-1 distance between the learned and ground-truth distribution of symmetries as a quantitative metric for evaluating symmetry discovery. Across all tested settings, such as $C_4$,  Tet and Oct groups, etc. LieFlow consistently achieves a lower W1 distance than LieGAN, indicating that our model more accurately recovers the underlying symmetries.
>
> We train both models on a single 4090. The per-epoch training time is 0.64s for LieGAN and 12.3s for LieFlow. The runtime difference is primarily due to the matrix logarithm, which LieFlow uses in its most general form to discover unknown symmetries without any prior assumptions. However, the matrix logarithm currently lacks an efficient GPU implementation. With GPU-optimized matrix logarithm routines, we expect comparable runtimes. Although symmetry-detection methods vary widely in assumptions and formulations, LieGAN is the most appropriate baseline for a generative-model–based approach, and the results demonstrate clear improvements of LieFlow over prior methods.
>
> References:
>
> [1] Yang, Jianke, Robin Walters, Nima Dehmamy, and Rose Yu. "Generative adversarial symmetry discovery." In International conference on machine learning, pp. 39488-39508. PMLR, 2023.
>
> 2. **Feasibility of SO(d) and GL(d)**
>
> We agree that our original expression may have unintentionally suggested empirical scalability, whereas our intended meaning was the generality of the framework, not the performance of the current implementation. We thank the reviewer for highlighting this point. LieFlow relies only on exponential/logarithmic maps and a left-invariant vector field parameterization, and therefore applies to any matrix Lie group such as $SO(d)$, $GL(d)$, or more structured subgroups without requiring architectural changes. In this sense, the method is not restricted to low-dimensional groups.
>
> At the same time, we fully acknowledge that scaling flow matching to large or non-compact groups introduces significant numerical instabilities (e.g., ill-conditioned geodesics, high-entropy priors). We have revised the text around line 186 accordingly to avoid overclaiming and clarify that the limitation lies in implementation-level stability, not in the conceptual scope of the framework. We have also added a note in the “Limitations and Future Work” section emphasizing the need to improve numerical scalability on high-dimensional groups.
>
> 3. **Discovering continuous symmetry**
>
> In the revision, we have added a 3D experiment demonstrating discovery of a continuous symmetry in the main text and Appendix C.2, specifically the subgroup $SO(2)\subset SO(3)$ corresponding to rotations around the z-axis. We apply LieFlow with $G=SO(3)$ on a point cloud whose ground-truth symmetry is an $SO(2)$ orbit, and visualize the results in Section 5.2 (Figure 5(d)) and Appendix C.2 (Figure 12(c), Figure 14, Figure 17). We thank the reviewer for raising this point.
>
> The learned distribution matches the ground-truth continuous symmetry closely: the Wasserstein-1 distance between the learned $p_\theta(g)$ and the true SO(2) distribution is $\textbf{0.03}$, showing that LieFlow can recover continuous symmetry without requiring discretization or manually enumerated modes.
>
> 4. **Method Writing**
>
> We thank the reviewer for pointing these out. For $x_0 = gx_1$, as $g$ is an element of the hypothesis group and not the data symmetry, $g$ acting on $x_1$ would give an element of the hypothesis group. For Algorithm 1, $A$ is the vector field that takes $x_0$ to $x_1$ and is the training target. Unlike standard conditional flow matching, we construct $x_0$ from $x_1$ using $g$ and because $G$ is a group, we can compute $A$ analytically as $A = \log(g^{-1})$. The model $v_\theta(x_t, t)$ is parameterized by the neural network and trained to model the vector field $A$. We have clarified these points in Section 4.2 of the revision.

---

> ### Author Response · Authors · 2025-12-03
> **Author Response [2/2]**
>
> 5. **Flow matching clarification**
>
> The reviewer is correct that in classical differential geometry, a "flow" strictly refers to the solution of a time-independent (autonomous) ODE satisfying the group property $\psi_{t+s} = \psi_t \circ \psi_s$, and that maps arising from time-dependent vector fields are more precisely called "evolution operators." We thank the reviewer for this correction. However, the machine learning literature on normalizing flows [1] and flow matching [2-4] has consistently adopted a broader usage of "flow." Rezende & Mohamed (2015) [1] define a flow as a sequence of invertible mappings and explicitly discuss infinitesimal flows governed by time-dependent dynamics. Similarly, the flow matching literature [2-4] defines flows as maps $\psi_t: \mathbb{R}^D \to \mathbb{R}^D$ transporting samples from $t=0$ to $t=1$, even when derived from a time-dependent velocity field $u_t$. Given that our work builds upon this literature, we follow its established conventions for consistency. That said, we appreciate the reviewer's suggestion and we have updated the description of flow matching in Section 2 to be more precise and also include a clarifying footnote noting that "$\psi_t$​ is technically an evolution operator arising from a time-dependent vector field; we follow standard terminology in the flow matching literature and use the term flow."
>
> References:
>
> [1] Rezende, D., & Mohamed, S. (2015). Variational inference with normalizing flows. ICML.
>
> [2] Lipman, Y., Chen, R. T., Ben-Hamu, H., Nickel, M., & Le, M. (2023). Flow Matching for Generative Modeling. ICLR.
>
> [3] Tong, A., Fatras, K., Malkin, N., Huguet, G., Zhang, Y., Rector-Brooks, J., ... & Bengio, Y. (2023). Improving and generalizing flow-based generative models with minibatch optimal transport. TMLR.
>
> [4] Lipman, Y., Havasi, M., Holderrieth, P., Shaul, N., Le, M., Karrer, B., ... & Gat, I. (2024). Flow matching guide and code. arXiv:2412.06264.
>
> 6. **Related work and Sub-optimal presentation**
>
> We have added the suggested related works (Shaw et al., 2024; Ko et al., 2024; Otto et al., 2023) to the Related Work section and included a brief discussion of how they relate to our approach. We have also corrected the typos and incomplete phrases noted by the reviewer, including the issues around lines 59–61, line 81, and line 161. Regarding the figure readability, we will adjust the font sizes and improve the presentation in the camera-ready version. We thank the reviewer for pointing out these issues.
>
> 7. **Discovering structured-preserving transformations**
>
> Our method is indeed compatible with structure-preserving (manifold-preserving) symmetry discovery, not only distribution-preserving transformations. We thank the reviewer for raising this conceptual question. LieFlow learns a distribution over group elements in $G$. If one is interested solely in the transformations that preserve the underlying data manifold, regardless of how the data are distributed on that manifold, then the support of the learned distribution provides the relevant symmetry structure. In other words, the method naturally separates which transformations preserve the manifold from how often they appear in the dataset.
>
> To illustrate this, we have added an experiment in Appendix C.5 where the $SO(2)$ transformation with Gaussian distribution is recovered. This demonstrates that LieFlow can discover structure-preserving symmetries without requiring the data to be uniformly distributed along the corresponding orbits.

---

### Official Review · Reviewer_gP36 · 2025-10-31

**Soundness:** 3
**Presentation:** 3
**Contribution:** 2
**Rating:** 4
**Confidence:** 2

**Summary:**

The paper proposes a method for learning subgroups of Lie groups from examples transformed by random elements of the subgroup. It uses flow matching on manifolds, the larger (superset) Lie group being the manifold in this case, to model the problem: The subgroup samples form a distribution on the larger (superset) Lie group, which can be identified by learning a corresponding generative model. When applying the method to shapes, it uses and requires (to my understanding) models with corresponding vertices which are geometrically identical up to transformations from the symmetry group and possibly small amounts of noise.
In experiments, the method is able to recover subgroups such as 4-fold reflection/rotation groups of larger sets of 2D and 3D transformations (SO(), GL(d)). The method begins to struggle with larger groups, such as octahedral discrete symmetry, where the dense coverage seems to lead to convergence problems. A final section gives an intuitively sound explanation in that attraction to local basins develops only slowly and accelerates quickly in the final training steps due to the initially (likely to be) symmetric attraction. This can be fixed by adapting the time scheduling of flow matching.

**Strengths:**

Discovery of symmetry and invariance in data is a very interesting and potentially fundamental problem in data science and machine learning; as such, I would see a high "prior likelihood" of impact for any progress on this matter (I could also say, I like the problem the paper is addressing).
The idea of trying to model the issue as just a density reconstruction / generative learning problem on a suitable space of parameters - transformations in a Lie group in this case - appears to be interesting and promising (although it seems that there has been some related work).
The concrete implementation using Lie groups and their Lie algebras for spanning the space and discovering regularity is quite elegant, and pulling the problem into the data domain by transforming the geometry accordingly seems to be the main technical idea needed to implement the rather abstract idea. This seems simple, straightforward and not too difficult to implement, all of which are in principle a big advantage.
I like the last part concerning the "last minute convergence" phenomenon (symmetric attraction being broken late in optimization) in particular: It does not only discover and honestly disclose one of the numerical issues facing the approach but also provides an interesting insight on when generative models in the style of flow-matching run into issues, which is probably interesting on its own as a case study of invariance being harmful to training, which might appear unintuitive at first sight. The paper also provides a strategy that can alleviate the issues.
Finally, I would also like to mention that I would think that the paper is very well written and enjoyable to read (it would, nonetheless, be useful to point out the main workings behind the formalism, which focuses mainly on the workings of the Lie-groups; in particular, one could discuss a bit more the geometry aspect, i.e., what kind of training data is expected and how data items are related to each other).

**Weaknesses:**

My main concern is that the method is only addressing a rather simple scenario: Having only one instance of the symmetric object given, which is replicated in symmetric copies with, as far as I understand, full vertex-to-vertex correspondence leads to a rather simple scenario. One could probably discover discrete symmetries by simple clustering (maybe meanshift in vertex-space) and continuous symmetry by shape matching (meaning here: computing relative transformations from corresponding vertices in the larger group) and applying PCA to the log-Matrices (in their local frame). I do not see where the full "power" of representation learning with deep networks is required or exploited. Or put differently: The method would probably not be able to discover that a randomly rotated sample of ModelNet-10 is frame invariant from looking at the examples, and, in my understanding, will certainly fail if there are not many copies of the same model included in the training data.
If true (I might misunderstand the approach or overinterpret the experiments - this would be a point to address in the discussion phase, in case), this does not mean that there is no value in the proposed ideas and analysis. I found the basic idea interesting, and the findings about the "last-minute" issues, which arise already in a simple scenario, quite insightful and potentially influential for building a more complex method that can handle strong invariance to geometry. I think that the paper makes an interesting point with valuable ideas but the overall contribution seems to be a bit too small for a top venue.

**Questions:**

Do you need (vertex-to-vertex) correspondences among the pieces of training data? In general, how much potential would there be for (more) invariance towards geometric variability mixed with varying transformations?

Also, in case I misunderstood the overall workings and idea, please correct me.

---

> ### Author Response · Authors · 2025-12-03
>
> We appreciate the reviewer's detailed and thoughtful analysis and clarify the concerns below.
>
> 1. **Vertex–to–vertex correspondence**
>
> LieFlow does not rely on vertex-wise alignment between point clouds. While we did not randomize point indexing in our experiments, our formulation does not require vertex-to-vertex correspondence. The flow matching predictor can be instantiated using permutation-invariant architectures (e.g., DeepSets), so that consistent vertex indexing is neither assumed or required.
>
> 2. **Why simple baselines (e.g., PCA in the Lie algebra) are insufficient.**
>
> The aim of our work is symmetry discovery in settings where orbit completeness does not hold. For example, MNIST with a random rotation applied independently to each digit, or any dataset where each sample appears at most under at most one unknown transformation. In such cases, there is no meaningful way to compute relative transforms between samples and thus PCA baselines cannot be applied. While such baselines may work in the simplified single-orbit scenarios, they fail in the more realistic setting where symmetry must be inferred from distribution-level consistency rather than explicit pairwise alignments. Furthermore, PCA would likely be unable to handle multiple mixed symmetries as the clustering would become combinatorially difficult, whereas our flow-based formulation would naturally handle these cases.
>
> 3. **Handling geometric variability across different shapes.**
>
> To address this concern, we added an experiment in Appendix C.4 where multiple geometrical shapes (irregular tetrahedron, triangular prism, cube, and octahedron) are mixed together, all augmented with the same tetrahedral symmetry. We set $SO(3)$ as the hypothesis group, and our method can successfully recover the tetrahedral subgroup from this heterogeneous dataset. This demonstrates that the method is not restricted to identical shapes or single-orbit settings.

---

### Official Review · Reviewer_iWYt · 2025-11-01

**Soundness:** 2
**Presentation:** 3
**Contribution:** 2
**Rating:** 2
**Confidence:** 4

**Summary:**

The authors propose training a flow matching model on Lie groups as a means of detecting symmetries within a dataset. Specifically, they train a flow matching model with data domain as a Lie group $G$, and assume the to-be-detected symmetry group $H$ is a subgroup of $G$, and symmetry is detected by examining if the learned distribution concentrates around a known symmetry group. To train via flow matching, LieFlow relies on computing exponential and logarithmic maps for interpolating training inputs and constructing training targets. The network is trained to predict tangent vectors (elements of the Lie algebra). Sampling relies on integrating the learned vector field with many small exponential maps. Experiments show that LieFlow can detect simple, low-order symmetries, but struggles with higher symmetry groups such as tetrahedral, octahedral, and icosahedral groups. The main challenge is found to be "last-minute mode convergence", which the authors analyze in detail, and partially overcome with a proposed power distribution timestep schedule.

**Strengths:**

To my knowledge, the idea of training a flow matching model as a means of detecting symmetries is novel.

The presentation of the background and LieFlow's formalism is elegant. The authors do a good job of contextualizing their work in the symmetry-detection literature.

Experiments are applied to novel toy problems, especially in learning a generative model on $GL(2, \mathbb{C})$.

The authors carefully analyze the "last-minute mode convergence" problem and provide insight with informative and visually appealing figures.

The paper proposes a power distribution time schedule to partially overcome the "last-minute mode convergence" problem.

**Weaknesses:**

I believe the direction is promising, but the paper could use more rounds of improvement.

A noticeable gap is that the paper does not make the connection of "last-minute mode convergence" with the pathologies of Riemannian flow matching (RFM) with geodesics on compact spaces. These pathologies are that the target vector field is discontinuous at the cut locus ([the surface/poles where geodesics meet](http://www2.mat.dtu.dk/info/mathematics/EmMa/cut_locus/clgif.html)), and the support of the probability path vanishes over time. This phenomenon has been reported in multiple works:
- Figures 2 & 3 in [1] demonstrate that RFM has difficulty learning complicated distributions on the 2D torus, echoed in Figure 4 of [2].
- Appendix G.2 of [3] shows that learning vector fields of geodesics on SO(3) is numerically unstable due to the sharp boundaries of the shrinking uniform distribution.
- Naive flow matching on the simplex results in learning a discontinuous vector field, while the support of the probability distribution vanishes over time. [4]

Figures 6&7 of LieFlow find similar conclusions to the above works, but the paper does not discuss these parallels. Had the authors been aware of this connection, they would know that Riemannian diffusion does not suffer from these pathologies. However, Riemannian diffusion is only briefly mentioned in the paper when citing [2], and [1] is not cited by the authors.

[1] Lou, A., Xu, M., Farris, A., & Ermon, S. (2023). Scaling Riemannian diffusion models. Advances in Neural Information Processing Systems, 36, 80291-80305.

[2] Yuchen Zhu, Tianrong Chen, Lingkai Kong, Evangelos A Theodorou, and Molei Tao. Trivialized momentum facilitates diffusion generative modeling on lie groups. arXiv preprint arXiv:2405.16381, 2024.

[3] Holderrieth, P., Havasi, M., Yim, J., Shaul, N., Gat, I., Jaakkola, T., ... & Lipman, Y. (2024). Generator matching: Generative modeling with arbitrary markov processes. arXiv preprint arXiv:2410.20587.

[4] Stark, H., Jing, B., Wang, C., Corso, G., Berger, B., Barzilay, R., & Jaakkola, T. (2024). Dirichlet flow matching with applications to dna sequence design. arXiv preprint arXiv:2402.05841.

Due to these pathologies, the performance of LieFlow is handicapped, limiting experiments to toy datasets and diminishing the impact of the work. On toy datasets, it is straightforward to find relative transformations between all data points and detect symmetries in that way. Because experiments are only performed on toy datasets, LieFlow is unable to demonstrate its potential advantages for detecting unknown symmetries in real-world data.

The actual detection of symmetry needs more explanation. It is easy to check if LieFlow's learned distribution concentrates around an intended symmetry group, but how will symmetries be detected when LieFlow is applied to real-world data, where the target symmetry group is a priori unknown?

The authors also need to explain how the framework of LieFlow will be able to detect soft symmetries in real-world data. The authors mention valency and electronegativity as symmetries in materials chemistry, as well as local symmetries in images, but it is unclear to me how LieFlow will be able to detect these symmetries.

There should be at least one experiment dedicated to an application with potential practical use, such as a real-world dataset of 3D molecules or point-clouds of 3D objects. One potential application is to rediscover torsional symmetries of 3D molecules.

**Questions:**

1. Given that Riemannian flow matching with geodesic paths on compact spaces suffers from pathologies, whereas Riemannian diffusion does not suffer from these problems, what is the justification for training a flow matching model rather than a diffusion model, especially in the context of recently developed efficient methods for Riemannian diffusion [1,2]?
2. How will soft symmetries, such as local image symmetries, or valency and electronegativity, be detected within the framework of LieFlow?
3. What is the motivation for detecting symmetries by training a generative model, versus detecting symmetries by directly checking whether the data satisfies transformations of certain groups?
4. Why is it necessary to choose a restricted hypothesis group? What stops us from starting with (if I understand correctly) the most general hypothesis group of $GL(n, \mathbb{C})$ for transforming $n$-dimensional data?
5. What is the simplest but practical real-world dataset that LieFlow could be tested on? What practical situations require detection of discrete groups?

nit-picking:
1. There's an extra "Text" element in Figure 1.
2. extra newline at line 60
3. line 413: "that C4 symmetry in clearly identified." - "in" should be "is"

[1] Yuchen Zhu, Tianrong Chen, Lingkai Kong, Evangelos A Theodorou, and Molei Tao. Trivialized momentum facilitates diffusion generative modeling on lie groups. arXiv preprint arXiv:2405.16381, 2024.

[2] Mangoubi, O., He, N., & Vishnoi, N. K. (2025). Efficient Diffusion Models for Symmetric Manifolds. arXiv preprint arXiv:2505.21640.

---

> ### Author Response · Authors · 2025-12-03
>
> We very much appreciate the reviewer's thoughtful and constructive feedback. We address each point in detail below.
>
> 1. **On the connection to Riemannian diffusion and geodesic-path pathologies**
>
> While Riemannian diffusion is a promising direction for future work, it is out of scope for the current paper. We appreciate the reviewer for pointing out the connection between “last-minute mode convergence’’ and known pathologies of Riemannian flow matching on compact manifolds. This is indeed an important perspective that we were unaware of, and we agree that recent work on Riemannian diffusion avoids several issues inherent to geodesic-based flow matching.
> While systematically comparing flow matching and diffusion models is currently outside the present scope of our work, we acknowledge that this is a promising future direction to discover Lie groups. We have added a discussion in the Limitations and Future Work section to explicitly highlight the relationship to Riemannian diffusion and note it as an important avenue for future exploration.
>
> 2. **On real-world application**
>
> While real-world experiments are an important next step, the core contribution of this paper is to establish and analyze a brand new and general framework for symmetry discovery on Lie groups. We agree that demonstrating LieFlow on real-world datasets would further strengthen the work. While our current experiments focus on controlled synthetic settings to investigate the behavior of flow matching on Lie groups, the framework itself does not rely on orbit completeness or explicit symmetry augmentation and can, in principle, be applied to real-world data.
> In fact, several practical domains contain exploitable discrete symmetries, such as 3D molecular torsions ($C_2$ or improper rotational symmetries),  or mechanical or CAD components with $ D_4 / C_4$ rotational symmetries. Evaluating LieFlow on these datasets is a natural next step, and we have added this to Section 6.1 Limitations and Future Work.
>
> 3. **On detecting soft or local symmetries**
>
> LieFlow in its current form is able to detect soft or local symmetries. As LieFlow learns a distribution over the hypothesis group, its ability to discover local symmetries directly depends on the hypothesis group. If the hypothesis group contains local symmetries (e.g. gauge transformations), LieFlow can in principle learn them. For the chemical examples, valency and electronegativity were used as global symmetries to reduce the search space for inorganic materials [1]. If such constraints are present in the dataset, LieFlow would be able to learn them as part of the underlying structure.
>
> Furthermore, because our method does not require the discovered symmetries to be a subgroup, it can also handle soft or approximate symmetries by learning only the group elements that appear in the dataset. We have shown this in Appendix C.5.
>
> References:
>
> [1] Davies, Daniel W., Keith T. Butler, Adam J. Jackson, Andrew Morris, Jarvist M. Frost, Jonathan M. Skelton, and Aron Walsh. "Computational screening of all stoichiometric inorganic materials." Chem 1, no. 4 (2016): 617-627.
>
> 4. **Why use generative modeling rather than directly checking invariances?**
>
> Directly testing whether $x$ and $gx$ lie in the same orbit requires explicit observations of transformed versions of the same data point, which are rarely available in real-world datasets. In contrast, LieFlow infers distribution-preserving transformations implicitly by modeling consistency across samples of the data distribution $q(x)$, without requiring paired or observations from the whole orbit. This enables symmetry discovery even when no transformed copies of the same object are provided. Furthermore, unlike GAN-based methods [1][2], LieFlow does not rely on adversarial training and can directly model distributions of multi-model transformation, which can improve models’ training stability and expressivity.
>
> [1] Yang, J., Walters, R., Dehmamy, N., & Yu, R. (2023, July). Generative adversarial symmetry discovery. In International conference on machine learning (pp. 39488-39508). PMLR.
>
> [2] Desai, K., Nachman, B., & Thaler, J. (2022). Symmetry discovery with deep learning. Physical Review D, 105(9), 096031.

---

> > ### Author Response · Authors · 2025-12-04
> >
> > 5. **Why use a restricted hypothesis group instead of the most general one?**
> >
> > Our framework is fully compatible with large hypothesis groups, including general matrix Lie groups. In practice, however, choosing a group that accurately reflects the types of realistic transformations seen in data like rotations or rigid motions can reduce ambiguity in the learned distribution and make the flow matching model easier to learn underlying symmetries. Extremely large non-compact groups like $GL(n,\mathbb{C})$ introduce ill-conditioned exponential/log maps (i.e., maps whose Jacobians vary rapidly in scale, leading to numerical instability, high sensitivity to perturbations, and difficulties in computing stable gradients) and high-entropy priors, which can make flow matching significantly more challenging. Thus, restricting the hypothesis group is not a limitation of the method but a practical choice for stability and interpretability.

---

### Author Response · Authors · 2025-12-03
**Executive Summary [1/2]**

We thank the Area chair for their time and consideration. Below is a summary of our main contributions and reviews, and our response to the reviews.

### **Executive Summary**

This work proposes LieFlow, a new method for symmetry discovery via flow matching on LIe groups. The key idea is to learn a distribution over a hypothesis group and learn a flow on the Lie algebra of the hypothesis group such that the learned map concentrates probability density on the true symmetries present in the data.

The main contributions of this paper are:
- Formulating symmetry discovery as flow matching on Lie groups
- Providing a unified framework for discovering continuous and discrete symmetries (including reflections via flow matching over the complex domain).
- Identifying and analyzing the fundamental challenge of ``last-minute mode convergence’’, where samples remain stationary until late in the flow, and introduce a new time schedule for symmetry discovery.


### **Updates in the Revised Paper**
- Added LieGAN baseline and new metric (Wasserstein-1 distance) to evaluate the quality of discovered symmetries. Our method outperforms LieGAN for all settings. (Reviewer cRXJ)
- New experiment to discover $SO(2)$: LieFlow correctly discovers $SO(2)$ symmetry from the $SO(3)$ hypothesis group (Reviewer cRXJ)
- New experiment with multiple different shapes: we successfully discover the correct tetrahedral symmetries (Reviewer gP36)
- New experiment to discover structure-preserving transformations / approximate symmetries: we can successfully learn a non-uniform distribution of group elements (Reviewer iWYt)
- Added connections to other related work, discuss Riemannian diffusion as future direction, clarifications on scalability and methodology.

---

### Author Response · Authors · 2025-12-03
**Executive Summary [2/2]**

### Reviewer Concerns and How We Addressed Them

| Reviewer | Main Concerns                                                                                                                                      | How We Address Them                                                                                                                                                                                                                                                                            |
|----------|----------------------------------------------------------------------------------------------------------------------------------------------------|------------------------------------------------------------------------------------------------------------------------------------------------------------------------------------------------------------------------------------------------------------------------------------------------|
| iWYt     | Connection of "last-minute mode convergence" with pathologies of Riemannian flow matching and how Riemannian diffusion does not have these issues. | Acknowledge this valuable connection and view this as an important future direction; added discussion in Section 6.1.                                                                                                                          |
| iWYt     | Limited to toy datasets, no real-world experiments                                                                                                 | Framework handles real-world data in principle; focus was on fundamentals. Added discussion in Section 6.1 |
| iWYt     | Detecting soft or local symmetries                                                                                                                 | LieFlow in its current form can detect soft or local symmetries, added experiment to learn non-uniform distributions of symmetries in Appendix C.5                                                                                                                                             |
| gP36     | Requires vertex-to-vertex correspondence, PCA might suffice.                                                                                       | We clarify the misunderstanding, method does not require this and is compatible with permutation-invariant architectures (e.g., DeepSets). PCA requires paired orbit samples.                                |
| gP36     | Limited to single-object scenarios                                                                                                                 | Added multi-object experiments with heterogeneous shapes (Appendix C.4)                                                                                                                                                                                                                        |
| cRXJ     | No comparison with existing methods                                                                                                                | Added LieGAN baseline with Wasserstein-1 metric; LieFlow outperforms across all settings (Tables 1-2)                                                                                                                                                                                          |
| cRXJ     | SO(d)/GL(d) feasibility claims not validated                                                                                                       | Acknowledge and revised text to clarify framework generality vs. empirical scalability to avoid overclaiming                                                                                                                                                                                             |
| cRXJ     | Writing unclear (x₀ "lives in G", Algorithm 1)                                                                                                     | Clarified in Section 4.2; added explanatory footnote                                                                                                                                                                                                                                          |
| cRXJ     | No continuous symmetry experiments                                                                                                                 | Added SO(2) discovery experiment with W1 distance of 0.03 (Section 5.2, Appendix C.2-C.3)                                                                                                                                                                                                       |

---

### Meta-Review · Area_Chair_ZH4e · 2026-01-07

**Summary:**

There are several major concerns raised by the reviewers:
- The methods is only experimented in synthetic data: while being theoretically sound, a clear limitation of the method is that it only works for synthetic data where the symmetry group is known by construction. In other settings, its application is limited.
- Generalization to soft- or local- symmetry: again, the generalization is only guaranteed when the soft-symmetry or local-symmetry is known during the construction of data.

**Reviewer Concerns:**

These two concerns is not fully addressed by the authors' claim. Without serious test on the real-world data, the method's application can be limited.

**Reviewer Scores:**

No change of score is expected.

---

### Decision · Program_Chairs · 2026-01-26

Reject